

**The interactive global fire module pyrE**
Keren Mezuman[1,2], Konstantinos Tsigaridis[1,2], Gregory Faluvegi[1,2], Susanne E. Bauer[2,1]
[1] Center for Climate Systems Research, Columbia University, New York, NY, USA
[2] NASA Goddard Institute for Space Studies, New York, NY, USA
*Correspondence to*: Susanne E. Bauer (Susanne.Bauer@columbia.edu)



**Abstract.** Fires affect the composition of the atmosphere and Earth's radiation balance by emitting a suite of reactive gases and particles. An interactive fire module in an Earth System Model (ESM) allows us to study the natural and anthropogenic drivers, feedbacks, and interactions of open fires. To do so, we have developed pyrE, the NASA GISS interactive fire emissions module. The pyrE module is driven by environmental variables like flammability and cloud-to-ground lightning, calculated by the GISS ModelE ESM, and parameterized anthropogenic impacts based on population density data. Fire emissions are generated from the actual flaming phase in pyrE (fire count), not the scar left behind (burned area), as is commonly done in other interactive fire modules. Using pyrE, we examine fire behavior, regional fire suppression, burned area, fire emissions, and how it all affects atmospheric composition. To do so, we evaluate pyrE by comparing it to satellite-based datasets of fire count, burned area, fire emissions, and aerosol optical depth (AOD). We demonstrate pyrE's ability to simulate the daily and seasonal cycles of open fires and resulting emissions. Our results indicate that interactive fire emissions are bias low by 32-42%, depending on emitted species, compared to the GFED4s inventory. The bias in emissions drives underestimation in column densities, which is diluted by natural and anthropogenic emissions sources and production and loss mechanisms. Yet, in terms of AOD, a simulation with interactive fire emissions performs just as well as a simulation with prescribed fire emissions.

**1 Introduction**

Open biomass burning (BB), the outdoor combustion of organic material in the form of vegetation, occurs on every continent, with the exception of Antarctica, at a scale observable from space. Open BB is perceived as a natural ecological process that has been modulating the carbon cycle for more than 420 million years [*Scott and Glasspool*, 2006]. However, in practice, BB has been mediated by human activities for more than 100,000 years [*Bowman et al.*, 2009, 2011; *Archibald et al.*, 2012]. *Bellouin et al.* (2008) estimated that at present, only about 20% of fires, compared to preindustrial times, are natural. *Andreae* (1991) estimated that in the tropics, where about 85% of fire emissions occur [*van der Werf et al.*, 2017], only 10% of fires are natural. In the USA, government records show that about 85% of fires are started by humans [*Balch et al.*, 2017]. Humans

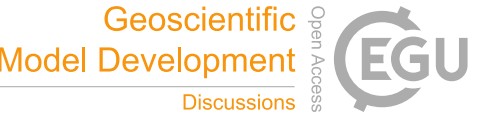

affect fires directly through ignition and suppression, and indirectly through man-made
changes to land surfaces and climate. According to *Hantson et al.* (2015), land-use
practices are the most important driver of human-fire interactions.

BB regimes are often classified based on ecosystem type like boreal, temperate,

and tropical forests, savanna and grassland, peat land, and agricultural fires [*Ichoku et al.*,
2012]. However, fire characteristics also vary between geographic regions of the same
ecosystem type; for example, boreal fires in Russia have very different intensity,
efficiency, and emissions than boreal fires in Canada [*Wooster and Zhang*, 2004]. *Ichoku*
*et al.* (2008) suggested an energy-based classification of open BB indicating fire intensity,
similar to hurricanes, using the radiative power of satellite-retrieved fires. Globally,
satellite retrievals show that on average about 350 Mha are burned annually [*Giglio et al.*,
2013; *Chuvieco et al.*, 2016], about 4% of the global vegetated area [*Randerson et al.*,
2012], an area similar to that of India. African fires contribute about 70% to the global
total burned area (BA), with about equal contributions from Northern Hemisphere Africa
(NHAF, Fig. 1) and Southern Hemisphere Africa (SHAF). The most flammable
ecosystem, globally and specifically in Africa, is the savanna [*Ichoku et al.*, 2008;
*Randerson et al.*, 2012; *Giglio et al.*, 2013], which in the tropics (23.5° N - 23.5° S) alone
is responsible for 62% (1341 TgC a$^{-1}$) of global carbon emissions (2200 TgC a$^{-1}$) [*van der*
*Werf et al.*, 2017]. Australian bushfires (grass and shrub) and South American savanna
fires are the third and fourth largest regional contributors, with BAs of about 50 Mha and
20 Mha annually, respectively. Globally, *Randerson et al.* (2012) estimated an additional
contribution of 120 Mha from small fires. The thermal anomalies used to identify those
fires, which are mostly associated with agricultural fires, are below the detection limit of
satellite-retrieved surface reflectance, and come with large uncertainties. Regionally,
small fires can have a significant contribution to BA. By adding the contribution of small
fires, burned area increases in Equatorial Asia (EQAS) by 157%, in Central America
(CEAM) by 143%, and in Southeast Asia (SEAS) by 90% [*Randerson et al.*, 2012]. This
highlights the regional importance of small agricultural fires to regional fire activity.
Forest fires, including small fires, contribute about 17 Mha annually to global BA, and
are dominant in Temperate North America (TENA), Boreal North America (BONA),
Boreal Asia (BOAS) and EQAS.





BB can exist when three conditions are met: fuel is available, fuel is combustible,
and ignition sources are present [*Schoennagel et al.*, 2004]. The coincidence of these
conditions is seasonal, making open BB an inherently seasonal phenomenon. The peak
month and duration of fire season are coupled to the seasonal cycle in precipitation,
especially in the tropics [*Giglio et al.*, 2006; *Hantson et al.*, 2017b]. In North America,
most fires occur over the plains of the Midwest and Southeast from early spring to
summer where they peak in June-July. Those anthropogenic fires are ignited as a mean of
agricultural land clearing. Similarly, around the summer months forest fires are common
along the Rocky Mountains, the Sierra Nevada mountain range, the Pacific Northwest,
and Boreal Canada and Alaska. Forest fires are either ignited on purpose, as part of forest
management practices [*Ryan et al.*, 2013], ignited by accident, as a by-product of the
expansion of urban life to the wildland interface [*Moritz et al.*, 2014; *Fischer et al.*, 2016;
*Radeloff et al.*, 2018], or ignited by lightning [*Díaz-Avalos et al.*, 2001]. In Central
America there is a south-to-north migration of fire activity, which follows the dry season.
Savanna burning in Colombia and Venezuela takes place between January-April,
followed by a May-August burning in Mexico. In South America most of the burning
takes place in the grasslands of southeast Brazil, set by ranchers for land management
practices, from June to mid-October [*Dwyer et al.*, 2000]. In Europe and Eurasia the BB
season is from April to September, with peaks in May, July and August. From April
through August, farmers in the breadbasket of Eurasia, from the Black Sea to Lake Baikal,
start fires to clear the land and burn crop residue. Siberian boreal fires, which are mostly
lightning-ignited, peak in July-August [*Dwyer et al.*, 2000]. Around the same time
Mediterranean fires peak. Trends in population density like land abandonment and shrub
encroachment, fuel the Mediterranean fires [*Butsic et al.*, 2015]. In NHAF the burning
season is from November to March, which peaks in December-January [*Giglio et al.*,
2013]. Then, the shift in the dry season to the Southern hemisphere dictates the SHAF
burning season from May to October, starting in the northwest and progressing to the
southeast [*Giglio et al.*, 2006]. Fires are mostly set on purpose to clear land of crop
residue and parasites, create firebreaks around settlements, and initiate regrowth of
vegetation [*Dwyer et al.*, 2000]. In SEAS the fire season, driven by land management,
starts in January and ends in early April, dictated by the monsoon circulation. BB in



eastern Asia, of mainly crop field residue, occurs between May-August. In EQAS
burning occurs between August and November. In Australia, most fires occur in the
grasslands of the Northern Territories, starting in the west and progressing to the east
from May to December. Additionally, fire activity occurs between January and March in
Southern Australia. The Southern Hemisphere BB activity is particularly sensitive to
natural modes of variability like El Niño Southern Oscillation (ENSO) [*Buchholz et al.*,
2018]. During an El Niño year regional BB emissions can be up to two times higher than
their regional average level, due to increased fire activity in tropical rainforests [*van der*
*Werf*, 2004; *Andela and Werf*, 2014; *Field et al.*, 2016; *Whitburn et al.*, 2016].

Although BB emissions have high spatiotemporal variability, their impact on
atmospheric composition is significant [*Crutzen et al.*, 1979; *Seiler and Crutzen*, 1980;
*Crutzen and Andreae*, 1990]. BB emissions impact air quality [*Johnston et al.*, 2012,
2014, 2016; *Bauer et al.*, 2019], and climate [*Ward et al.*, 2012; *Lasslop et al.*, 2019].
Emitted pollutants include ozone precursors like methane (~49 Tg a$^{-1}$), carbon monoxide
(~820 Tg a$^{-1}$), and NO$_x$ (mostly emitted as NO, ~19 Tg a$^{-1}$) [*Andreae*, 2019]; the latter
two are also deleterious for health on their own. In addition to gaseous pollutants, BB
emits particulate matter (a total of ~85 Tg a$^{-1}$) like primary emitted black carbon (~5 Tg
a$^{-1}$) and organic carbon (~36 Tg a$^{-1}$), as well as precursors of brown carbon, and
secondary organic and inorganic aerosols like non-methane volatile organic compounds
(NMVOC, ~58 Tg a$^{-1}$), ammonia (~9.9 Tg a$^{-1}$), sulfur dioxide (~6 Tg a$^{-1}$), and NO$_x$
[*Andreae*, 2019]. Exposure to these pollutants at high concentrations or for a long period
of time can compromise the cardiorespiratory system and lead to death [*Lelieveld et al.*,
2015]. These pollutants, along with BB-emitted greenhouse gases (GHGs) like carbon
dioxide (CO$_2$; ~13,900 Tg a$^{-1}$) and nitrous oxide (N$_2$O; ~1.38 Tg a$^{-1}$), interact with
radiation, directly and indirectly. Fires are a net source of carbon dioxide only where
vegetation regrowth is inhibited, i.e. in deforested areas; otherwise BB is not viewed as a
source of CO$_2$ but as "fast respiration" [*van der Werf et al.*, 2017]. Absorbing black and
brown carbon [*Lack et al.*, 2012; *Lack and Langridge*, 2013; *Laskin et al.*, 2015], and
reflecting primary and secondary organic and inorganic aerosols interact with solar
radiation directly by scattering and absorbing radiation, and indirectly by modifying
clouds. The radiative properties of particles and their hygroscopicity are also influenced





by their mixing state [*Bauer and Menon*, 2012]. For example, when black carbon (BC) is
coated it becomes even more absorbing per unit mass [*Bond and Bergstrom*, 2006]. There
is evidence that smoke plumes can suppress or invigorate precipitation [*Feingold et al.*,
2001; *Andreae et al.*, 2004; *Tosca et al.*, 2015]. Aerosols impact cloud height and cover
by modifying the heat profile of the atmosphere and increasing the number of cloud
condensation nuclei. There are large uncertainties associated with aerosols' impact on
climate. Modeling studies suggest that the aerosol effects from BB emissions overrides
the BB-GHG effect to a net negative radiative forcing [*Mao et al.*, 2013], with the
indirect effect of clouds dominating the forcing [*Ward et al.*, 2012]. The present day BB
forcing is estimated at -0.5-(-0.1)±0.05 Wm$^{-2}$ [ Ward *et al.*, 2012; *Mao et al.*, 2013; *Jiang*
*et al.*, 2016; *Landry and Matthews*, 2016; *Lasslop et al.*, 2019].

The quantification of speciated BB emissions is challenging due to the fact that no

one fire is the same as another [*Ito and Penner*, 2005]. The composition of the resulting
smoke plume depends on the fuel type, burning conditions (i.e. flaming or smoldering),
fuel consumption, and on background chemistry. More complete combustion has a higher
fraction of oxidized species (e.g. $CO_2$ and $NO_x$) while smoldering fires release more
reduced species (e.g. CO, $NH_3$, NMVOCs). Thus, emissions in different regions
contribute different amounts of pollutants; Indonesia, for example, is responsible for 8%
of global carbon BB emissions, but 23% of methane BB emissions [*van der Werf et al.*,
2017]. Emissions are sensitive to season and region. Even within one region, like a boreal
forest, emissions from crown fires differ from those from ground fires. The amount of
fuel consumed by a fire is highly variable and depends on fuel load, density, moisture,
vegetation type, and on environmental factors such as wind speed, soil moisture and soil
composition. Additional challenges relate to external forcing like insect herbivory,
mammal grazing, and manmade land fragmentation and deforestation [*Schultz et al.*,
2008]. The quantification of BB emissions has an even bigger importance during
preindustrial times, where fire emission are identified as the largest source of uncertainty
for aerosol loading in Earth system models [*Hamilton et al.*, 2018]. BB emissions are a
key quantity needed for quantifying the unperturbed-from-humans background conditions
of the atmosphere [*Carslaw et al.*, 2013].





Traditionally, fires are included in climate models using emission inventories
[*Lamarque et al.*, 2010; *van der Werf et al.*, 2010, 2017; *van Marle et al.*, 2017]. Some
models have the ability to simulate BB emissions interactively with a varying level of
complexity [*Thonicke et al.*, 2001; *Arora and Boer*, 2005; *Pechony and Shindell*, 2009; *Li*
*et al.*, 2012; *Lasslop et al.*, 2014; *Hantson et al.*, 2016; *Mangeon et al.*, 2016; *Rabin et al.*,
2017; *Zou et al.*, 2019]. On the one end of the spectrum, there are statistically-based
models, and on the other end there are detailed empirical and physical process-based
models. Statistical models are skilled at making predictions based on present-day
relationships between climate and fire (their training data). Process-based models
encapsulate the complex feedbacks within the climate system at various levels. They
combine physical processes such as fuel condition, cloud-to-ground lightning ignitions,
and wind-driven fire expansion. Some models also include simplified empirical
relationships of anthropogenic ignition and suppression, which, at present, are not
understood in a dynamic process level. Though less accurate than observational datasets,
when trying to simulate individual fire events, fire models provide the unique advantage
of linking the atmosphere, biosphere and hydrosphere in a consistent way, a crucial step
when studying Earth System interactions. They are also able to predict fire during climate
periods for which we have no observational data available (e.g. preindustrial and future).
State-of-the-art process-based fire models are well equipped to study the
feedbacks between the climate system and fires [*Hantson et al.*, 2016]. However, there is
indication that they lack accurate predictive capabilities, as they only partly capture
trends in present day observations. For example, satellite products show a global decrease
in burned area from about 500 Mha $a^{-1}$ in 1997 to 400 Mha $a^{-1}$ in 2013, a trend which fire
models do not capture [*Andela et al.*, 2017]. This trend is mostly driven by land
fragmentation and grazing practices over African savanna, highlighting the challenge of
fire models to account for the combined changes in climate, vegetation and socio-
economic drivers [*Forkel et al.*, 2019].
In this paper we present a new global fire module, pyrE, based on an improved
scheme of [*Pechony and Shindell*, 2009, 2010] with new, state-of-the-art, capabilities.
The pyrE module is process-based, as it includes the two basic parameters of fuel
availability and combustibility, which are used to calculate fire count. It utilizes empirical





relationships with population density to account for the anthropogenic impact on fire
ignition and suppression. However, unlike other fire models where fire suppression is
applied uniformly across all regions [*Rabin et al.*, 2017], in pyrE fire suppression
depends both on population density and region. Additionally, pyrE uses fire count to
derive emissions, and is therefore more directly connected to the actual fires, in contrast
to other fire models that use BA, a measure more indicative of fire's effect on the
landscape. The fire module is part of the NASA GISS ModelE Earth System model,
ModelE2.1 (an updated version based on *Schmidt et al.* (2014)), and is described below.
**2 Model description**
pyrE, from the Greek word for fire (pyr), is a global fire module within GISS
ModelE. It incorporates the fire count parameterization of *Pechony and Shindell* (2009,
2010), with the addition of fire spread and BA, following the Community Land Model's
(CLM) approach [*Li et al.*, 2012]. The module is a collection of physical processes like
flammability, natural and accidental ignition, suppression, fire spread, and fire emissions
(Fig. 2). The climate model input required, includes surface temperature, surface relative
humidity (RH), precipitation, surface wind speed, vegetation density and type, cloud-to-
ground lightning frequency and population density. Like many fire modules it lacks
explicit intentional ignition (e.g. crop, deforestation) and peat fires.
**2.1 Flammability**
Flammability is a parameter that indicates conditions favorable for fire occurrence
[*Pechony and Shindell*, 2009, 2010]. It is a unit-less number that ranges between zero and
one, and is calculated using vapor pressure deficit $(VPD)$, monthly-accumulated
precipitation, and vegetation density $(VD)$.
$VPD$, an indicator of drought [*Seager et al.*, 2015; *Williams et al.*, 2015], is
calculated via the Goff-Gratch equation [*Goff and Gratch*, 1946; *Goff*, 1957] using the
saturation vapor pressure $(e_s)$ and surface relative humidity $(RH)$:
$$VPD = e_s \left(1 - \frac{RH}{100}\right) \quad (1)$$
Where $e_{st} = 1013.245 \, [mb]$ is the saturation vapor pressure at the boiling point
of water and $e_s = e_{st} 10^{Z(T)}$ depends on temperature $(T)$:
$$Z(T) = a\left(\frac{T_s}{T} - 1\right) + b \cdot \log\left(\frac{T_s}{T}\right) + c\left(10^{d\left(1 - \frac{T_s}{T}\right)} - 1\right) + f\left(10^{h\left(\frac{T_s}{T} - 1\right)} - 1\right) \quad (2)$$


With the coefficients: $a = -7.90298; b = 5.02808; c = -1.3816 \cdot 10^{-7}; d =$

$11.344; f = 8.1328 \cdot 10^{-3}; h = -3.49149$ [*Goff and Gratch*, 1946], and $T_s =$
$373.16$ [°$K$] (water boiling point temperature).

The precipitation dependence of flammability is in the form of an inverse

exponential (Following [*Keetch and Byram*, 1968]):

$f(R) = \exp(-c_R R)$ (3)

Where $R$ is the surface rain rate in mm per day and $c_R = 2$ [$day/mm$] is an

empirical constant [*Pechony and Shindell*, 2009].

Vegetation density ($VD$) is taken as the normalized leaf area index (LAI) in the

land fraction of a grid cell, varying between 0 for no vegetation and 1 for dense
vegetation.

We modified the original calculation proposed by [*Pechony and Shindell*, 2009]

by calculating flammability only for the fraction of the model's grid cell that is not
burned from previous fires. The flammability $F$ at a time step $t$ in a grid cell $(i, j)$ is:

$F(t) = 10^{Z(T(t)_{i,j})} \left(1 - \frac{RH(t)_{i,j}}{100}\right) VD(t)_{i,j} \left(1 - \frac{BA(t)_{ij}}{LA_{i,j}}\right) \exp(-c_R R(t)_{i,j})$ (4)

Where $LA_{i,j}$ is the total land area (LA) in the grid cell $(i, j)$.

**2.2 Ignition**

Natural and anthropogenic ignition varies in space and time, and is necessary for

the calculation of fire count. If ignition is zero, the resulting fire count will be zero,
independent of flammability. Natural ignition is in the form of cloud-to-ground lightning
frequency, which is interactively calculated in ModelE2.1 [*Price and Rind*, 1992, 1993].
The parameterization of anthropogenic ignition follows *Venevsky et al.* (2002) and is
based on the assumption that in sparsely populated regions people interact more with the
natural environment, thus increasing the potential for ignition. The parameterization uses
population density data and empirical scaling factors, as described by *Pechony and*
*Shindell* (2009), and does not include intentional ignition. The number of anthropogenic
accidental ignitions per km$^2$ per month is:

$I_A = k(PD)PD\alpha$ (5)

Where PD is the population density; $k(PD) = 6.8PD^{-0.6}$ represents the varying

anthropogenic ignition potentials as a function of population density; $\alpha = 0.03$ is the



number of potential ignitions per person per month. Coefficients are taken following
*Pechony and Shindell* (2009) and *Mangeon et al.* (2016) which utilized correlation
calculations done by *Venevsky et al.* (2002).
**2.3 Suppression**
A first-order approximation of the impact of population density on explicit fire
suppression was proposed by *Pechony and Shindell* (2009). According to that
parameterization, more fires are suppressed in densely populated areas compared to
sparsely populated areas, regardless of ignition source. Specifically, suppression varies
from 5% to 95% of fires. However, fire management is a region-specific practice, which
depends on cultural norms and economic capabilities. For example, fire suppression in
the United States of America (USA) is much more aggressive than most regions in the
world. In the Middle East, vegetation is sparse and is mostly near centers of human
population for agricultural purposes. Natural ignition is almost inexistent and most fires
are controlled by human activities, which make the impact of suppression stronger. Fire
suppression for open BB is not commonly practiced in most parts of Africa. In some
regions of Africa, fires are used as a tool to clear land for agriculture and to prevent
savanna overgrowth and the spread of pests. Hence, we improved the simplistic approach
suggested by *Pechony and Shindell* (2009), guided by the results presented in Sect. 5.1.1.
We use the complement of the fraction of suppressed fires that is the fraction of non-
suppressed fires, $f_{NS}$:
$$f_{NS} = \begin{cases} 0.2\exp{(-0.05PD)}, & USA\ and\ MIDE \\ 1, & Africa \\ 0.05 + 0.9\exp{(-0.05PD)}, & Elsewhere \end{cases} \quad (6)$$

Similarly to *Pechony and Shindell* (2009), constant values are selected in a
heuristic manner, due to the lack of appropriate global data.
**2.4 Fire count**
Fire count is a key metric as it is used to drive burned area and fire emissions in
pyrE. The number of fires in a time step per km$^2$ is calculated as the product of
flammability, sum of natural and anthropogenic ignition, and suppression [*Pechony and*
*Shindell*, 2009] (Fig. 2):
$$N_{fire}(t)_{i,j} = F(t)_{i,j} \cdot \left( I_N(t)_{i,j} + I_A(t)_{i,j} \right) \cdot f_{NS}(t)_{i,j} \quad (7)$$

**2.5 Burned area (BA)**

We adopted the process-based approach of *Li et al.* (2012) to calculate fire spread and burned area. The burned area in grid cell $(i, j)$ at a model time step $t$ is the product of fire count and the weighted average over plant functional types (PFTs) of the area burned by one fire:

$$BA_{i,j} = N_{fire}(t)_{i,j} \cdot \sum_v a_{i,j,v} \cdot f_{i,j,v} \quad (8)$$

Where $f_{i,j,v}$ is the fractional area covered by plant functional type v, and the burned area of a single fire $a_{i,j,v}$ is assumed to have an elliptical shape (Fig. 3). Wind speed, surface relative humidity, and vegetation type control the eccentricity of the ellipsoid that represents the burned area of a single fire (based on *van Wagner* (1969)):

$$a_{i,j,v} = \frac{\pi ROS^2 \tau^3}{4LB} \left(1 + \frac{1}{HB}\right)^2 \quad (9)$$

Where $ROS$ is the rate of fire spread, $LB$ is the length-to-breadth ratio, and $HB$ is the head-to-breadth ratio. The stronger the wind, the more eccentric the ellipse, i.e. the bigger the length-to-breadth ratio:

$$LB = 1 + 10 \cdot (1 - \exp(-0.06W)) \quad (10)$$

Where $W$ is the surface wind speed in m s[-1].

Strong winds also increase the head to back ratio; the ratio of the downwind spread compared to the upwind spread:

$$HB = \frac{LB + \sqrt{LB^2 - 1}}{LB - \sqrt{LB^2 - 1}} \quad (11)$$

The rate of spread (ROS) of a fire is a function of vegetation type, wind speed, and atmospheric and soil moisture:

$$ROS = ROS_{max} \cdot gW \cdot f_{RH} \cdot f_\theta \quad (12)$$

$ROS_{max}$ is the maximum fire spread rate. Following *Li et al.* (2012), we set it to 0.2 m s[-1] for grasses, 0.17 m s[-1] for shrubs, 0.15 m s[-1] for needle leaf trees, and 0.11 m s[-1] for other trees. *Li et al.* (2012) estimated the fire spread coefficients to be on the lower range of observed ROS, but are yet higher than the global value of 0.13 m s[-1] suggested by *Arora and Boer* (2005).

The limit of the fire spread is set by:

$$gW = \frac{2L_B}{1 + \frac{1}{H_B}} g0 \quad (13)$$



Where $g0 = \frac{1+HB_{max}^{-1}}{2LB_{max}} \approx 0.05$

$f_{RH}, f_\theta$ are the dependencies of fire spread on RH and root zone soil moisture:

$$f_{RH} = \begin{cases} 1 & RH \leq RH_{low} \\ \frac{RH_{up}-RH}{RH_{up}-RH_{low}} & RH_{low} < RH < RH_{up} \\ 0 & RH \geq RH_{up} \end{cases} \quad (14)$$

Following *Li et al.* (2012), we set $RH_{low} = 30\%, RH_{up} = 70\%$ *and* $f_\theta = 0.5$ as

ModelE2.1 does not simulate prognostic root zone soil moisture.
**2.6 Emissions**

Trace gas and aerosol emissions are calculated using PFT (denoted by v) and

chemical specie (s) specific emission factors $(EF_{s,v})$. The emissions per grid cell $(i,j)$ of
specie $s$ at a model time step $t$ are calculated by:

$E_{i,j,s}(t) = N_{fire}(t)_{i,j} \cdot \sum_v EF_{s,v} \cdot f_{i,j,v}$ (15)

Where $E_{i,j,s}(t)$ is the emissions flux rate in kg m$^{-2}$ s$^{-1}$, $N_{fire}(t)_{i,j}$ are the fire

count, $EF_{s,v}$ are the offline emission factors, and $f_v$ is the fractional area of that PFT in
the grid cell.

Emission factors describe the PFT-specific speciated mass (in kg) of the smoke,

normalized per fire (Table 1). Emission factors were calculated offline using ModelE2.1
PFTs, annual mean global MODIS Terra fire count, and GFED4s emissions from the
period of 2003-2009. Our technique, known as multivariate curve fitting, matched the
emissions within the PFT fraction of the grid cell with the respective fire count. We
correlated GFED4s emissions with MODIS fire count as a function of the fraction of
modeled PFTs in a grid cell and calculated different emission factors per PFT.
**2.7 Implementation within ModelE**

ModelE2.1 can be used with either GFED4s prescribed fire emissions or

interactive pyrE emissions. The pyrE module generates emissions at every model time
step with ESM-simulated climate as a driver. Flammability is calculated only in the
fraction of grid cells with natural vegetation. It is driven by the simulated surface RH,
surface temperature, monthly accumulated precipitation, and LAI. LAI is calculated by
Ent [*Kim et al.*, 2015], the Terrestrial Biosphere Model component of ModelE2.1, and is
currently derived from 2005 MODIS LAI data [*Tian et al.*, 2002a, 2002b]. Cloud-to-





ground lightning, calculated by ModelE2.1, is used as the natural ignition source. Most
ESMs have low skill in reproducing flash rate distributions [*Murray*, 2016], and the GISS
model is no exception. A qualitative comparison with the World Wide Lightning
Location Network (WWLN) (not presented here) showed that modeled cloud-to-ground
lightning, which makes up only about 30% of total lightning, is bias-high in ModelE2.1.
We decided to use a simple scaling factor of 0.1 in the calculation of natural ignition to
better match observed flash rates, as improving the lightning parameterization is beyond
the scope of this study. Gridded population density (PD) that drives both anthropogenic
ignition and fire suppression is based on historical data for years prior to 2010 [*Klein
Goldewijk et al.*, 2010], and on future projections (not used in this study) for years past
2010. PD has a time resolution of 10 years and is interpolated in between.

The modeling approach presented in this paper provides a good reproduction of
the seasonality compared to satellite retrievals (see Results section). However, the
simulated magnitude of fire count and burned area was too small compared to satellite
retrievals and required the use of a scaling factor, a common practice among other fire
models [*Pfeifer et al.*, 2013; *Knorr et al.*, 2014; *Hantson et al.*, 2016; *Mangeon et al.*,
2016; *Zou et al.*, 2019]. To calibrate the global modeled fire count to MODIS retrievals,
we used a global scaling factor of 30 for all fire count. A similar approach was taken by
*Pechony and Shindell* (2009). We scaled burned area by a factor of 250 to reach the
magnitude of GFED4s. Nevertheless, even with this large correction factor, burned area
has a very minor impact on fire count and fire emissions as it accounts for a small
fraction of the grid cell that is able to burn.

**3 Model configuration**

We used ModelE2.1 with a spatial resolution of 2° in latitude by 2.5° in longitude,
40 vertical layers and a model top at 0.1 hPa. The vegetation component of ModelE2.1 is
the Ent Terrestrial Biosphere Model (Ent TBM), which is coupled with the land use/land
cover data in the model [*Kim et al.*, 2015]. Ent prescribes leaf area index (LAI) for 14
plant functional types (presented in Table 1) derived from MODIS 2005 data (cover and
biome types [*Friedl et al.*, 2010]; LAI [*Tian et al.*, 2002a, 2002b]), historical crop cover
[*Pongratz et al.*, 2008], and vegetation heights from [*Simard et al.*, 2011].



In this study we show results from runs of ModelE2.1 coupled to the aerosol
microphysical scheme MATRIX (Multiconfiguration Aerosol TRacker of mIXing state)
[*Bauer et al.*, 2008]. MATRIX simulates aerosol formation, condensation and
coagulation, calculates the size distribution of aerosols and tracks their mixing state. Sea
salt, dust, and dimethyl sulfide (DMS) emissions were calculated interactively, driven by
the simulated climate, while other natural and anthropogenic fluxes, except for fires, were
prescribed from the CEDS (Community Emissions Data System) inventory [*Hoesly et al.*,
2018].
In the following, we will present a simulation with pyrE turned on, generating
interactive fire emissions, and a simulation with pyrE turned off, using prescribed 2005
climatological GFED4s emissions instead. Also, we will discuss sensitivity studies using
two simulations where pyrE generates interactive fire emissions but suppression is
changed from a global parameterization to a regional one. Prescribed climatological
monthly varying mean (1996-2004) sea surface temperature and sea ice thickness and
extent were used as boundary conditions [*Rayner et al.*, 2003].
**4 Reference datasets**
The data below are based on a composite of level 3 Aqua and Terra Moderate-
resolution Imaging Spectro-radiometer (MODIS) Collection 5.1 data [*Giglio et al.*,
2003b; *Giglio*, 2013], unless otherwise stated. Aqua and Terra are sun-synchronous, near-
polar orbiting satellites with a global continuous record of more than 15 years; Aqua was
launched in May 2002 and Terra in December 1999. Aqua's overpass time is 1:30AM
and 1:30PM local, and Terra's overpass time 10:30AM and 10:30PM local, and their
period is between one to two days. All reference data used in this study are interpolated
and re-gridded to the resolution of ModelE2.1.
**4.1 Fire count**
To detect fires, MODIS uses brightness temperatures (thermal anomaly) derived
from two channels. Channel 31, that saturates at 400° K, and either channel 21, that
saturates at 500° K, or channel 22, that saturates at 331° K. Channel 22 is preferred over
21 as it has a higher signal to noise ratio, but when it saturates, or has missing data,
channel 21 is used [*Justice et al.*, 2002; *Giglio et al.*, 2006].





In our study we used the monthly cloud-corrected fire count (CloudCorrFirePix)
climate model grid data (MYD14CMH, MOD14CMH). The spatial resolution of the data
is 0.5°. Static, persistent hot spots are excluded from this product [*Giglio*, 2013]. Because
of its non-uniform spatial and temporal sampling, raw MODIS data are biased high at
high latitudes [*Giglio et al.*, 2003a, 2006]. The product we used is corrected for the
multiple satellite overpasses, the missing data, and variable cloud cover. Cloud cover
hinders MODIS retrievals. The fire count in the product we used is normalized to the
fraction of cloud cover in a pixel. In highly cloudy pixels, the product is set to zero. The
local time of retrieval matters for fire detection, as fires are driven by the daily cycle in
solar heating. The largest number of fire count is detected during daytime, with an order
of magnitude difference between daytime fire count detections and nighttime fire count
detections [*Ichoku et al.*, 2008]. Thus, differences are evident between the Aqua and
Terra retrievals. This motivated us to use data from the two satellites in our analysis. We
calculate and utilize climatological monthly means from the period 2003-2016.
**4.2 Burned area**
We used burned area from the Global Fire Emissions Database (GFED) version
4s that includes small fires [*van der Werf et al.*, 2010, 2017; *Randerson et al.*, 2012;
*Giglio et al.*, 2013]. The GFED4s inventory is based on multi-sensor MODIS data,
involving both reflectance and thermal anomalies measurements from Aqua and Terra.
MODIS detects burned area using the 650 nm, 1200 nm, and 2100 nm reflectance bands.
Retrievals must be free from cloud contamination and free from active fires within the
500 m MODIS grid cell. First, to generate the GFED4s data, MODIS burned area
collection 5.1 data (MCD64A1 product) are aggregated to a 0.25° grid. Then, burned area
from small fires is added. The burned area of small fires is statistically estimated using
active fire count detected by MODIS (a composite of both Aqua and Terra). Both the
ratio and correction factor are estimated each year as a function of region, season, and
vegetation type [*Randerson et al.*, 2012; *van der Werf et al.*, 2017]. Due to the projection
of the MODIS reflectance product over the thermal anomaly one, some resampling errors
occur. To partially correct this error, region-specific factors ranging from 0.88 in Africa
to 1.12 in boreal Asia are applied. In this study we use climatological monthly means of
burned area from the period 2003-2016.



**4.3 Biomass burning emission inventory**

GFED4s emissions are derived from the multiplication of burned area and fuel consumption [*van der Werf et al.*, 2010, 2017]. As such, they have the same spatial and temporal resolution as burned area, of 0.25° by 0.25° and a month. Fuel consumption is calculated using an estimation of fuel loss and combustion completeness, which are calculated using MODIS-based metrics such as differences in normalized burned area (dNBR), normalized vegetation index (NDVI), and land surface temperature (LST). These metrics inform about changes in green vegetation, canopy and soil water, and landscape charring. The satellite-based data are used as input to the Carnegie–Ames–Stanford Approach (CASA) biogeochemical model [*Randerson et al.*, 1996] to calculate the dry matter burned. Then, emission factors [*Andreae and Merlet*, 2001; *Akagi et al.*, 2011] are applied to convert the dry matter burned to PFT-specific speciated gas and aerosol phase emissions. *Kaiser et al.* (2012) and *Pan et al.* (2019) showed that there are regional biases in older and current versions of GFED; being especially biased low in the Southern Hemisphere compared to AERONET aerosol optical depth (AOD). In order to eliminate the strong interannual BB variability, our analysis used GFED4s mean climatological data of 1995-2010.

**4.4 Fire regions**

The analysis we present below is based on the widely used fire regions (Fig. 1) as defined by GFED [*Giglio et al.*, 2006; *van der Werf et al.*, 2006]. The regions are defined based on climate and fire regimes, and are widely used as basis regions for global fire studies.

**4.5 Aerosol optical depth**

The impact of fire emissions on atmospheric composition is investigated by comparing monthly Aqua and Terra MODIS retrievals of AOD at 550nm [*Remer et al.*, 2005; *Platnick et al.*, 2015]. AOD describes the entire atmospheric column-integrated extinction of aerosols. MODIS AOD data are a useful tool in the study of simulated BB plumes [*Voulgarakis and Field*, 2015; *Johnson et al.*, 2016; *Bauer et al.*, 2019]. The AOD data we used has a 1° spatial resolution. The monthly mean data (MYD08_M3 and MOD08_M3 products) have been averaged over the period 2003–2007 to create monthly climatologies centered around the year 2005. The AOD product we use includes





improvements made via the Dark Target algorithm [*Kaufman et al.*, 1997], which was
developed particularly for retrievals over dark vegetated surfaces [*Wei et al.*, 2019].
However, the algorithm fails at retrieving valid AOD data over bright surfaces like desert
areas [*Levy et al.*, 2013], which we discard. Here we use collection 6.1 data.
**5 Results and discussion**
**5.1 Fire count**
**5.1.1 Regional suppression**
First we want to demonstrate how the parameterization with regionally-dependent
fire suppression improves the simulation of fire count compared to the original simplified
global fire suppression proposed by *Pechony and Shindell* (2009) (Fig. 4). Our goal was
to improve the fire parameterization in regions where the seasonality was captured in
timing but not in magnitude. We propose regional modifications to Africa (NHAF,
SHAF), a region that drives global fire activity, and had a distinct mismatch in fire count
compared to satellite retrievals. Originally, over NHAF the fire seasonality was too flat,
while over SHAF it matched MODIS-Terra, but was orders of magnitude smaller than
MODIS-Aqua. Since fire suppression for open BB is not commonly practiced in rural
Africa, eliminating it over NHAF and SHAF helped resolve the seasonal cycle (Fig. 4
and Eq. 6). The two other regions we modified are TENA and Middle East (MIDE). Over
both of those regions the simulated fire seasonality was too strong. Increasing fire
suppression over MIDE and TENA greatly improved our simulations compared to
MODIS retrievals.
The pyrE module is skilled at capturing the fire seasonality in regions identified
by *Forkel et al.* (2017) as controlled by temperature and wetness (climate controls), like
Southern Hemisphere South America (SHSA) (Fig. A1). However, there are regions that
our parameterization does not simulate well, mainly due to the fact that the fire activity
there is driven by land use practices and intentional fire ignitions, which pyrE does not
resolve. For example, in TENA we are missing the spring peak of agricultural fires.
Similarly, over Europe and Boreal Asia (Fig. A1) we are missing the winter and spring
fires associated with intentional ignition [*Dwyer et al.*, 2000; *Ganteaume et al.*, 2013].
Other regions where the seasonality is not well captured, likely due to the fact that it is
driven by intentional ignitions, include Central America, Northern Hemisphere South



America, Central Asia, Southeast Asia, and Equatorial Asia. Over Australia, the model captures neither the magnitude nor the timing of the BB seasonality. This is in part due to the model's poor performance of the simulated cloud-to-ground lightning ignitions in that region (not shown).

In all simulations going forward we used the regional suppression scheme.

**5.1.2 Daily cycle**

We looked at the fire count daily cycle to see if it can explain the differences between Aqua, Terra, and the model. The monthly mean fire count detected by Aqua and Terra is expected to be different due to their different overpass times. In Fig. 5, pyrE simulates a distinct daily cycle in fire count in different locations. The simulated daily cycle is most strongly controlled by the simulated daily cycle in flammability (not presented here), matching the daily solar cycle. pyrE's ability to resolve a daily cycle of fire activity highlights the dynamic nature of a process-based fire model.

Using 30-minute simulation output, we sampled all surface grid cells at the daytime overpass time of MODIS Terra, 10:30am local time, and MODIS Aqua, 1:30pm local time. We focused on the daytime overpass time of Terra and Aqua since about 95% of fire count detections occur then [*Ichoku et al.*, 2008]. Our results in Fig. 6 and Fig. 7 indicate that, globally, simulated fire count sampled at daytime overpass is bias-high compared to MODIS retrievals from the respective satellite, for much of the year. On a global annual mean, the model sampled in daytime Terra overpass time is higher than MODIS Terra fire count by 45%, while the model sampled in daytime Aqua overpass time is higher than MODIS Aqua fire count by 13%. However, this behavior differs by region and maximizes in NH sub-Saharan Africa and SH central Africa. The simulated fire count is bias-low compared to MODIS retrievals along the coast of west Africa, in eastern southeast Asia and Australia. The implications of these findings are that even though the simulated monthly mean fire count is in the range of Terra and Aqua (Fig. 4, A1), the simulated fire count is in fact higher than MODIS retrievals. Considering that the actual number of fire count is likely higher than the number retrieved by MODIS, as cloud contamination is decreasing its detection efficiency, it is conceivable that a model weakly high-biased compare to the satellite retrievals is realistic. All results presented





later were not sampled according to a satellite overpass time, but instead were averaged
over the whole length of the day.



**5.2 Burned area**

The simulated burned area is bias-low compared to the GFED4s inventory (Fig. 8,

A2). The total annual simulated burned area (10-year climatological mean) is 31.5 Mha
while GFED4s burned area (mean of 2003-2016) is 38.1 Mha. However, this behavior is
region-specific. The simulated burned area is lower compared to GFED4s over northern
hemisphere Africa, particularly in November-December, over central and equatorial Asia,
and over Australia. The simulated burned area (Fig. 8, A2) reflects the spatial distribution
and seasonality of simulated fire count (Fig. 8, A1). GFED4s burned area and MODIS
fire count do not always have the same seasonality, for example during October-
December. During this season the satellite-retrieved fires produce a higher burned area
relative to other seasons. The fire activity driving this behavior occurs in the savanna of
sub-Saharan Africa, and northern hemisphere South America. In those regions and times
of the year the normalized mean bias of modeled burned area is at least twice the size of
the normalized mean bias of fire count, e.g. in NHAF a bias of 6.5 for burned area and 1-
3 for fire count, depending on the MODIS satellite. This implies that for every fire
modeled in these regions and season a smaller area is simulated to burn compared to the
reference datasets.

Why is the burned area per fire relationship in simulations much weaker than it is

in the reference datasets? Two contributing factors are: prescribed PFT and simulated
wind. The prescribed PFT distribution present in the model is rudimentary; it is
comprised of 11 flammable vegetation types (Table 1). As for surface winds, the
simulated wind patterns driving burned area are averaged over a coarse grid cell
(2°x2.5°). Simulated wind does not represent sub-grid scale processes and is not fueled
by the fire's energy, which is likely contributing to an underestimation of the spread of
burned area. However, though wind directly impacts burned area, it does not play a major
role in the distribution of simulated fires, since burned area itself has a minor impact on





fires due to its small percentage in a grid cell. At most burned area reaches less than 18%
of the naturally vegetated fraction of a grid cell, and is on average less than 1%.
**5.3 Emissions**

Due to the intricate processes involved in burned area spread, most fire models

struggle to reproduce the observed trend [*Andela et al.*, 2017] and seasonality [*Hantson et*
*al.*, 2017a] of burned area. A more direct approach would be to use fire count, similar to
the approach of *Pechony and Shindell* (2009, 2010) and *Pechony et al.* (2013).

The main source regions for fire emissions are NHAF, EQAS, SHSA, and SHAF.

Emissions are well simulated over SHSA and SHAF (Fig. A3-A5), both in terms of
timing of the seasonality and in magnitude. The main regions where simulated emissions
are lower than GFED4s are sub-Saharan Africa and Indonesia (Fig. 8). However, more
generally, simulated gaseous and particulate emissions are globally bias low compared to
GFED4s emissions (Table 2). This behavior is most prominent in sub-Saharan east Africa
and in EQAS, mainly in Indonesia (Fig. 8). To a lesser degree, simulated fire emissions
are also weaker compared to GFED4s in the boreal regions (Fig. A3-A5). The
contribution from these regions to the global total is an order of magnitude smaller
compared to the main source regions.

The weaker emissions compared to GFED4s are responding to the following inputs:

offline emissions factors, lack of crop and peat fires, LAI, and prescribed PFTs. The
emission factors that generate fire emissions are derived using multivariate statistical
analysis. Though we used seven full years (2003-2009) of data to derive the factors, it
might have generated biases in emissions. Areas that burn annually are properly sampled,
but areas that have a fire cycle that is longer than a seven year might be biased high or
low, depending on whether they were included in the training dataset or not. Also, crop
and peat fires are not explicitly included in the simulated emissions, as intentional
ignition is not parameterized in pyrE. Specifically, fires are not applied to the crop faction
of a grid cell, and peat surfaces are not included in the PFTs. However, our method of
deriving the offline emission factors uses MODIS fire count and GFED4s emissions, and
does not distinguish between intentional and accidental fires. Hence, intentional fires are
indirectly accounted for in the global sum. However, this indirect inclusion of intentional
fires does not necessarily add missing fire emissions in the correct locations. The LAI in



Ent, ModelE's DGVM, is based on 2005 MODIS retrievals. Though we cannot estimate
the role that the lack of interactive LAI plays, it is certainly not optimal, neither for fire
count simulation, nor for fire emissions that are derived from these fire count. Unlike fire
count, fire emissions are strongly tied to the map of PFTs. The offline emission factors
are based on prescribed PFTs, and the interactive emissions themselves are applied
according to the sub-grid PFT distribution. The prescribed PFT distribution present in the
model might be different than reality, and those differences affect emissions. In the
model, the PFTs in areas where emissions are bias high compared to GFED4s there is a
high percentage (>50%) of the following PFTs: evergreen broadleaf trees (Amazon,
central Africa), cold broadleaf trees (northeast America, Europe), and drought broadleaf
trees (central Africa and northern India). In EQAS, a region with bias low simulated
emissions, close to 100% of the prescribed PFTs is evergreen broadleaf trees, which in
reality is replaced by crops. The bias-low emissions in EQAS are very likely tied to the
lack of prescribed peat PFT. In areas with bias low emissions modeled PFTs are mainly
(>50%) c4 grass (sub-Saharan Africa, Australia), deciduous needle leaf trees (boreal
regions), and arid shrubs (S Africa, Australia).
**5.4 Composition**
**5.4.1 Column load**
In order to quantify how the model skill changes with the inclusion of pyrE
instead of prescribed emission inventory data in ModelE2.1, we compare a simulation
with interactive fires to a simulation with prescribed BB sources. Though emissions are
mostly bias-low compared to GFED4s, this behavior is less evident in the column density
(Fig. 9). For most BB emitted species, the simulation with interactive fires has lower
column densities than the simulation with prescribed emissions (Table 2) with a bias
ranging from -6.3-0.5% for gaseous species, -4.8% for black carbon and -16% for organic
aerosol. However, the column densities are only partly driven by fire emissions, as those
make up less than 35% of total global emissions of either CO, organic aerosol, and black
carbon emissions. Non-emissions production-and-loss mechanisms also impact column
densities.
The difference in column densities between the two simulations is greatest over
north sub-Saharan Africa, Indonesia, and the boreal regions. The behavior is region-



specific, and some regions like central Africa and northern hemisphere South America
have higher column densities compared to the simulation with prescribed emissions. The
differences between the two simulations are more prominent for organic aerosol than any
of the other species (Fig. 9, Table 2), while the differences in the spatial distribution of
CO are marginal.
**5.4.2 Aerosol optical depth (AOD)**

In Fig. 10 we compare climatologically-simulated clear-sky AOD with MODIS

AOD (Aqua) for January, April, July, and October. The conclusions from Terra products
are similar to Aqua's, and will not be presented here, for brevity. In a regional
perspective, simulated AOD is able to reproduce the seasonality and spatial distribution
of MODIS-retrieved pollution over west and central Africa, east and southeast Asia, and
the Arabian sea. The simulations of ModelE2.1 has higher AOD compared to MODIS
over the tropical eastern Pacific, an artifact due to the model's skill in simulating
stratocumulus cloud decks, which have been improved in a newer version of the ESM
(ModelE3).

Model performance as a function of interactive versus offline fire emissions is

similar in terms of AOD (Fig. 11). Both simulations have persistently lower (0-30%)
AODs over central Africa and central South America compared to MODIS. The locations
with an outstanding difference in performance between the simulations are in central sub-
Saharan Africa in January and July, and over a small area in Indonesia (Kalimantan)
during October. In January over central sub-Saharan Africa the simulation with pyrE has
AOD values (NHAF regional mean AOD of 0.26) closer to MODIS (NHAF regional
mean AOD of 0.2) than a simulation with prescribed fire emissions (NHAF regional
mean AOD of 0.33).while in July it is the simulation with pyrE (NHAF regional mean
AOD of 0.53) that is more bias high than the prescribed one (NHAF regional mean AOD
of 0.46). Over EQAS in October the simulation with prescribed fires has an AOD of
~0.28 while the simulation with pyrE has an AOD of ~0.18. AOD in this region is
sensitive to peat fires, which are not included in ModelE, strongly impacting pyrE's
results. Globally, mean AOD simulated with interactive fire emissions is 0.142 while
mean AOD simulated with prescribed fire emissions is 0.146. The fact that pyrE has a
marginal performance in climatological runs when compared against a simulation using





the more accurate offline emissions is a strong indication that it is a robust module that
can be used with confidence at time periods where offline emissions are not available.
Finally, we demonstrate the contribution of BB emissions to total clear-sky AOD
by comparing the simulations with both prescribed and interactive fire emissions to a
simulation that has no fire emissions at all (Fig. 12). In the simulation with prescribed fire
emissions, clear sky AOD is on average 10% higher than it is in a simulation with no fire
emissions. In a simulation with pyre clear sky AOD is about 7.5% higher than it is in a
simulation with no fire emissions. The impact of BB emissions on AOD is most
pronounced in the source regions of Africa and the Amazon. In those regions the
difference in AOD varies between 0.15-0.3. It is important to note that the differences in
AOD are not only due to impact of BB emissions, but also reflect climate variability,
which impacts aerosol lifetime and interactive dust emissions.
**6 Conclusions**
The development of pyrE, allowed us for the first time to interactively simulate
climate and fire activity with GISS-ModelE2.1. The pyrE module, which is based on a
the fire parameterizations of *Pechony and Shindell* (2009), was expanded to include fire
spread and burned area, following the approach of *Li et al.* (2012). This study set out to
simulate the climatology of fires, and not individual fire events. Like only a few other fire
models [*Zou et al.*, 2019], pyrE was developed with consideration of regional behavior.
The new fire suppression scheme depends on population density, but also on geographic
regions. The new scheme reflects more intense fire suppression in the USA and Middle
East, and revokes fire suppression in Africa, which improved the fire count seasonality
simulated by pyrE compared to satellite retrievals. Fire count seasonality is well
simulated in the fire source regions: the Amazon, SH Africa, and NH Africa, with the
exception of being bias-low compared to MODIS during November-December. This is
due to the lack in parameterization of intentional ignitions and agricultural fires.
The regional model skill of fire count was also demonstrated in the simulated
burned area. Burned area in southern hemisphere Africa was well simulated by the model,
while less active fire regions like temperate and boreal North America, Boreal Asia
Europe, and Middle were bias high compared to GFED4s. Other regions like Australia,
sub-Saharan and West Africa in November-December, Central Asia and Southeast Asia





in January-March were bias low. Though the seasonality of simulated burned area
reflects that of simulated fire count, the bias of burned area compared to GFED4s data is
at least double that of fire count. Burned area is a quantity that most fire models struggle
with. Wind speed, a driver of burned area, is averaged over a coarse grid cell, with no
feedback from fire heat and energy, which can be a contributing factor to the lower
simulated burned area values. The prescribed rudimentary PFTs of the model are a
simplified version of the real world and thus can be a source of additional uncertainty.
Finally, the rate of spread of burned area, a function of the burning vegetation type, that
pyrE and other fire models use is on the lower end of field observations. A higher rate of
spread could help to both override the scaling factor used for burned area, and to reduce
the negative bias compared to GFED4s.
Unlike other fire models, fire emissions in pyrE are driven directly by fires
instead of burned area. Emissions are based on online fire count calculations and offline
emission factors derived as described in Sect. 2.6. In contrast to the fact that simulated
fire count are bias-high compared to MODIS, globally, fire emissions are bias-low
compared to GFED4s. Fire emissions are well-simulated over the southern hemisphere
with the exception of Australia. Emissions are bias low over the northern hemisphere
including northern sub-Sahara, with the exception of NH South America, which is bias
high. The bias of fire count compared to MODIS in Australia and in sub-Saharan Africa
during November-December propagates to emissions. The emission factors, which were
calculated offline using MODIS fire count and GFED4s fire emissions and were applied
based on the prescribed PFTs of the model, have their own limitations. They are based on
a training dataset of seven years, which would introduce biases in regions where fire
cycle is longer than seven years. Also, they rely on the modeled PFTs, enhancing the
emissions dependency on the prescribed PFT and the lack of peat. Emission factors do
not distinguish between intentional and accidental fires, thus they indirectly account for
all fire emissions, which reduce existing biases, although the regional distribution of them
will not match the locations of intentional fires, unless natural vegetation burning occurs
in the vicinity.
Less emissions compared to GFED4s means lower column densities and lower
AOD when comparing a simulation with interactive fires to one with prescribed fires.





However, as these quantities depend on climate feedbacks including processes other than
fire, e.g. additional emission sources, precipitation, deposition, transport, and chemistry,
the differences between the two simulations dilute. Nonetheless, a comparison with
MODIS AOD demonstrates that AOD from a simulation with interactive fire emissions is
comparable to AOD from a simulation with prescribed fire emissions.

The work presented here highlights that timing matters just as much as magnitude.

This is true for fire distribution, emissions, and atmospheric composition. Timing is also
the reason why intentional ignition was excluded from pyrE. Intentional ignition, namely
land clearing and agricultural fires, depends on region and crop specific planting and
harvesting times. To include it would require crop functionality in ModelE, which was
not present during the time of our development. Further future development should focus
on the inclusion of intentional ignition and agricultural fires which are seasonal in nature,
derived from crop planting and land clearing times. This addition could perhaps improve
model performance over regions like equatorial Asia, Southeast Asia, and Central
America as well as override the global scaling factors applied to fire count and burned
area. The use of scaling factors is a common practice among fire models, and should be
carefully and transparently documented. Also, enhancing the prescribed PFTs, especially
via the addition of peat is imperative when studying fires. Peat exists as well outside of
tropical Asia. There are immense reservoirs of peat in Africa [*Dargie et al*., 2017], as
well as the boreal regions [Yu, 2012], where it used to be trapped under permafrost. Peat
will likely become an even bigger source of fire emissions in the future. Improvement of
the cloud to ground lightning parameterization may also prove useful, as changes to
natural ignition will likely have significant impacts on Australian and boreal fire
emissions. Finally, almost no fire models include fire energy. However, given that the
heat component of fires interact with the climate system, and can also be used to derive
more accurate emissions (as demonstrated by *Ichoku and Ellison* (2014)), it is worthwhile
taking it into consideration.






## 7 Code availability

pyrE is in line with state-of-the-art fire models, and can be easily applied to other ESMs. Information on ModelE, including access to online data and descriptions are available at http://www.giss.nasa.gov/tools/modelE. The pyrE module is included in ModelE version 2.1. The source code, along with documentation, can be downloaded from the NASA Goddard Institute of Space Studies website: https://simplex.giss.nasa.gov/snapshots/.

**Acknowledgements.** Climate modeling at GISS is supported by the NASA Modeling, Analysis, and Prediction program. The authors acknowledge funding from NASA's Atmospheric Composition Modeling and Analysis Program (ACMAP), contract NNX15AE36G. Resources supporting this work were provided by the NASA High-End Computing (HEC) Program through the NASA Center for Climate Simulation (NCCS) at Goddard Space Flight Center.

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





**Tables**
Table 1 Fire emission factors for the different plant functional types (PFTs) in ModelE2.1.
Factors are in units of kg per fire per PFT in the grid cell. For organic and black carbon
units kg is substituted with kg of carbon.

| PFT | CO | $NO_x$ | $SO_2$ | $NH_3$ | Alkenes | Paraffin | OC | BC |
|---|---|---|---|---|---|---|---|---|
| Cold Broadleaf Deciduous | 113392 | 1529 | 555 | 2101 | 106 | 69.8 | 3437 | 767 |
| Needle leaf | 481485 | 1559 | 4168 | 10722 | 422 | 373 | 36753 | 1844 |
| Drought Broadleaf | 230829 | 4835 | 1687 | 2340 | 214 | 108 | 10667 | 1382 |
| Evergreen Broadleaf | 249906 | 4905 | 1438 | 2847 | 220 | 102 | 10941 | 1434 |
| Evergreen Needle leaf | 146622 | 1197 | 972 | 2277 | 137 | 89.1 | 6537 | 821 |
| Cold Shrub | 105936 | 241 | 878 | 2006 | 104 | 72.1 | 6562 | 357 |
| Arid Shrub | 39268 | 1009 | 262 | 378 | 36.6 | 18.5 | 1479 | 238 |
| C3 Annual Grass | 26761 | 690 | 147 | 313 | 25.1 | 13.9 | 728 | 173 |
| C3 Arctic Grass | 251702 | 1094 | 2315 | 5065 | 489 | 226 | 15551 | 1159 |
| C3 Perennial Grass | 41043 | 908 | 270 | 438 | 38.8 | 20.7 | 1504 | 257 |
| C4 Grass | 117577 | 3152 | 795 | 1196 | 110 | 57 | 4339 | 726 |













Table 2: Total fire emissions and global mean column loads of fire emitted species.

| Species | Variable | pyrE | GFED4s | Bias [%] |
|---|---|---|---|---|
| CO | Emissions [Tg a$^{-1}$] | 2.14E+02 | 3.51E+02 | -39 |
| | Column Load [kg m$^{-2}$] | 7.22E-04 | 7.71E-04 | -6.3 |
| OA | Emissions [TgC a$^{-1}$] | 1.31E+01 | 2.29E+01 | -42 |
| | Column Load [kg m$^{-2}$] | 8.52E-07 | 1.02E-06 | -16 |
| BC | Emissions [TgC a$^{-1}$] | 1.25E+00 | 1.84E+00 | -32 |
| | Column Load [kg m$^{-2}$] | 7.25E-09 | 7.62E-09 | -4.8 |
| NO$_x$ | Emissions [Tg a$^{-1}$] | 4.27E+00 | 6.76E+00 | -36 |
| | Column Load [kg m$^{-2}$] | 5.94E-07 | 5.91E-07 | 0.5 |
| NH$_3$ | Emissions [Tg a$^{-1}$] | 2.43E+00 | 4.15E+00 | -41 |
| | Column Load [kg m$^{-2}$] | 2.15E-07 | 2.23E-07 | -3.5 |
| SO$_2$ | Emissions [Tg a$^{-1}$] | 1.34E+00 | 2.25E+00 | -40 |
| | Column Load [kg m$^{-2}$] | 2.67E-06 | 2.69E-06 | -0.7 |
| Alkenes | Emissions [Tg a$^{-1}$] | 1.94E-01 | 3.18E-01 | -39 |
| | Column Load [kg m$^{-2}$] | 5.73E-08 | 5.70E-08 | 0.5 |
| Paraffin | Emissions [Tg a$^{-1}$] | 9.79E-02 | 1.65E-01 | -40 |
| | Column Load [kg m$^{-2}$] | 2.36E-07 | 2.42E-07 | -2.4 |
















**FIGURES**

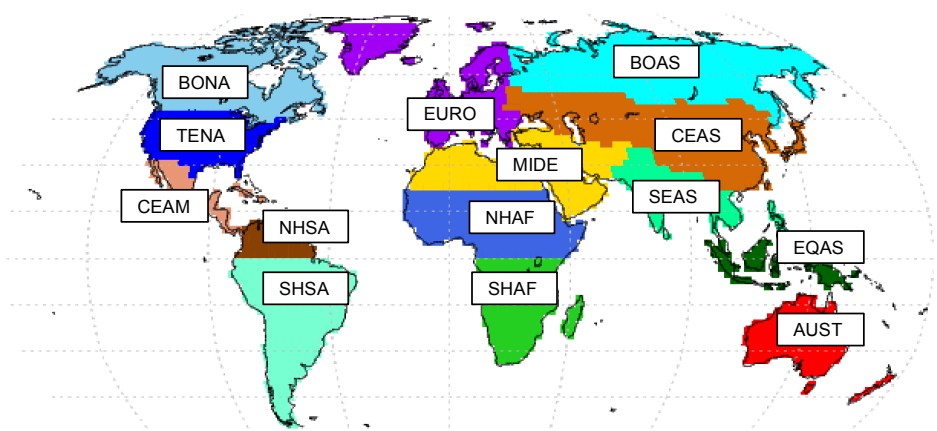


| | |
|---|---|
| BONA Boreal North America | NHAF Northern Hemisphere Africa |
| TENA Temperate North America | SHAF Southern Hemisphere Africa |
| CEAM Central America | BOAS Boreal Asia |
| NHSA Northern Hemisphere South America | CEAS Central Asia |
| SHSA Southern Hemisphere South America | SEAS Southeast Asia |
| EURO Europe | EQAS Equatorial Asia |
| MIDE Middle East | AUST Australia and New Zealand |


Figure 1. GFED basis regions regrided to the resolution of ModelE2.1 of 2° in latitude by
2.5° in longitude.












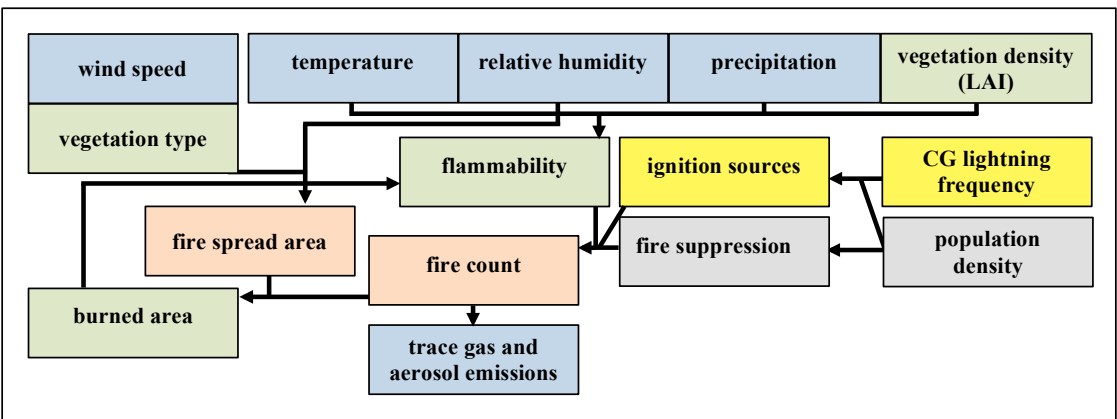



Figure 2. Structure of the fire parameterization of pyrE. Processes related to atmospheric
properties in blue, surface properties in green, ignition and suppression in yellow and
gray, and fire properties in red.




















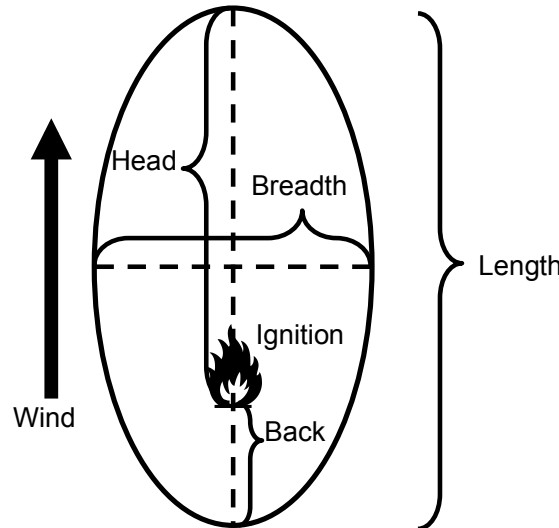



Figure 3. Approximation of a single fire spread. Based on *van Wagner* (1969) and *Arora*
*and Boer* (2005).



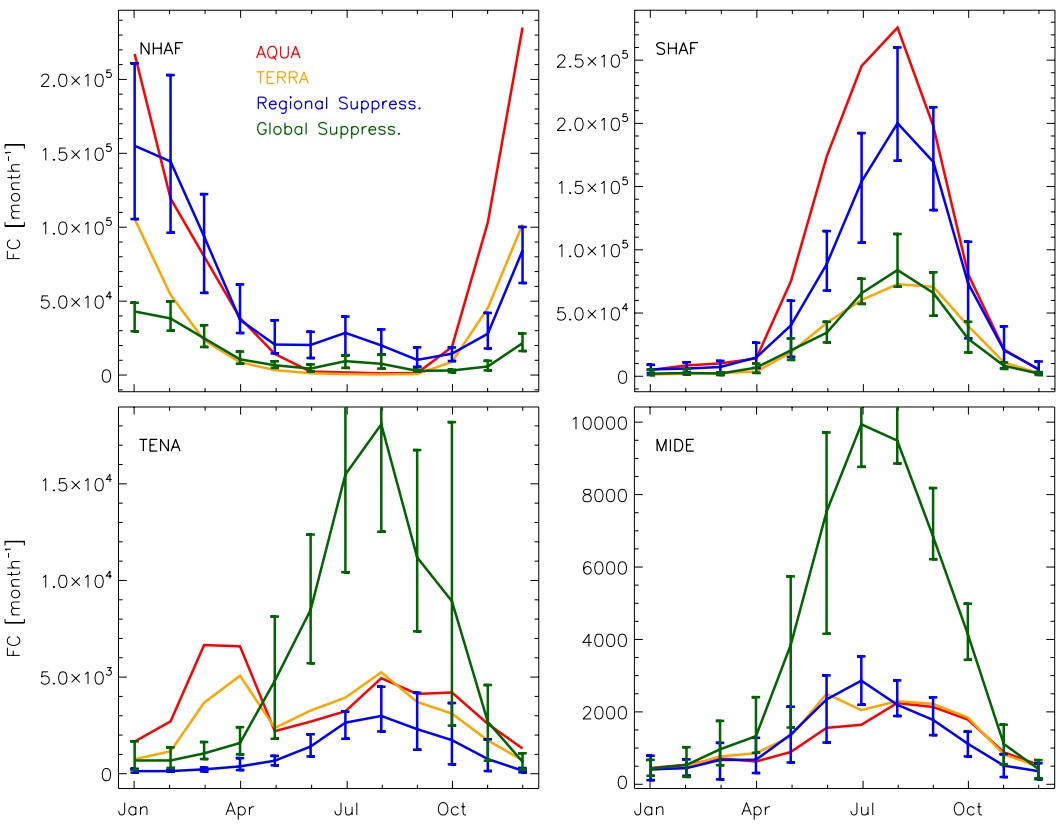

Figure 4: Seasonality of total fire count for NHAF (top left), SHAF (top right), TENA (bottom left) and MIDE (bottom right) observed by MODIS Aqua (red) and Terra (orange) and simulated with explicit regional suppression (blue) and generic global suppression parameterization (green); Eq. 6. Error bars represent the range over 10-year climatological simulations. Note that TERRA and AQUA have different overpass times, and the model data presented here are monthly means. Also, note the different scale in each panel.



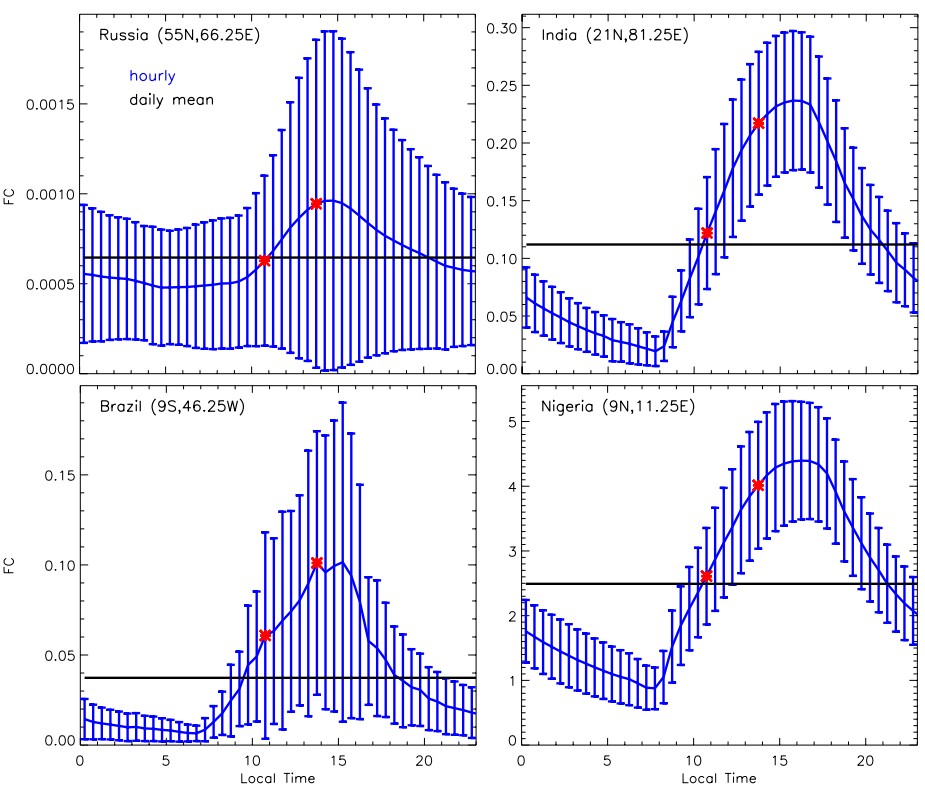

Figure 5: Daily mean cycle in fire count (FC, blue line) and daily mean (black line) at 4 locations during the month of January. The daytime overpass times of Terra (10:30am) and Aqua (13:30pm) are marked with a red star. Error bars represent the range during the month. Note the different scale in each panel.

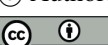



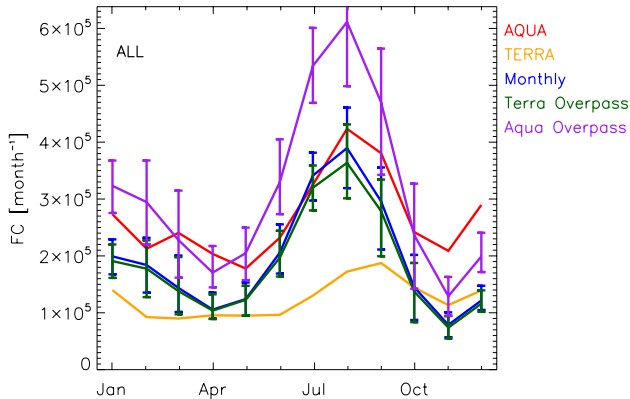


Figure 6: Global seasonality of total fire count (FC) by MODIS Aqua (red) and Terra
(orange) and simulated by the model: monthly mean (blue), monthly mean sampled at the
daytime Terra overpass time (green), and sampled at the daytime Aqua overpass time
(purple). Error bars represent the 10-year range in the simulation.









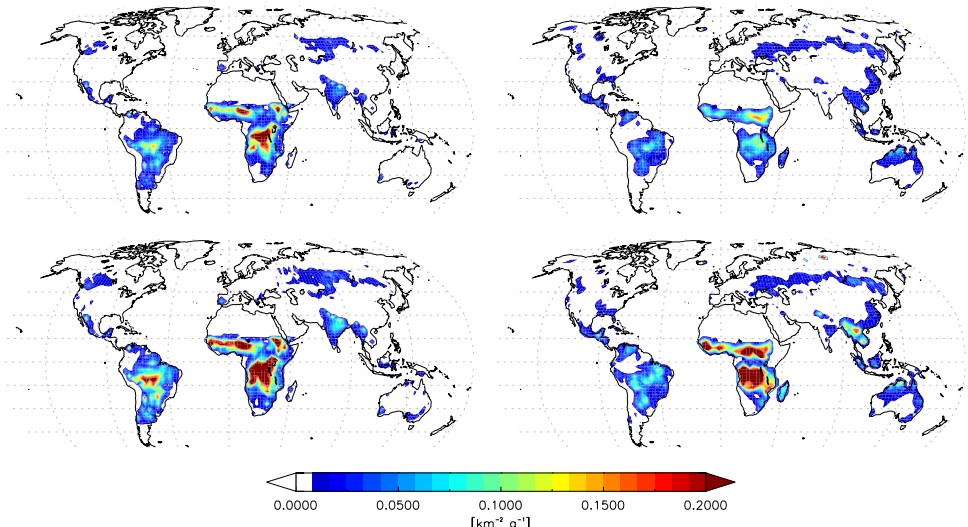


Figure 7: Annual mean model (left) and MODIS (right) fire count. Modeled annual mean

is based on an ensemble of 10 simulations. Simulated fires sampled at the daytime Terra

overpass time, 10:30am local time (upper left) and daytime Aqua overpass time, 1:30pm

local time (lower left). MODIS fire count is based on MODIS Terra (upper right) and

MODIS Aqua (lower right) from 2003-2016.



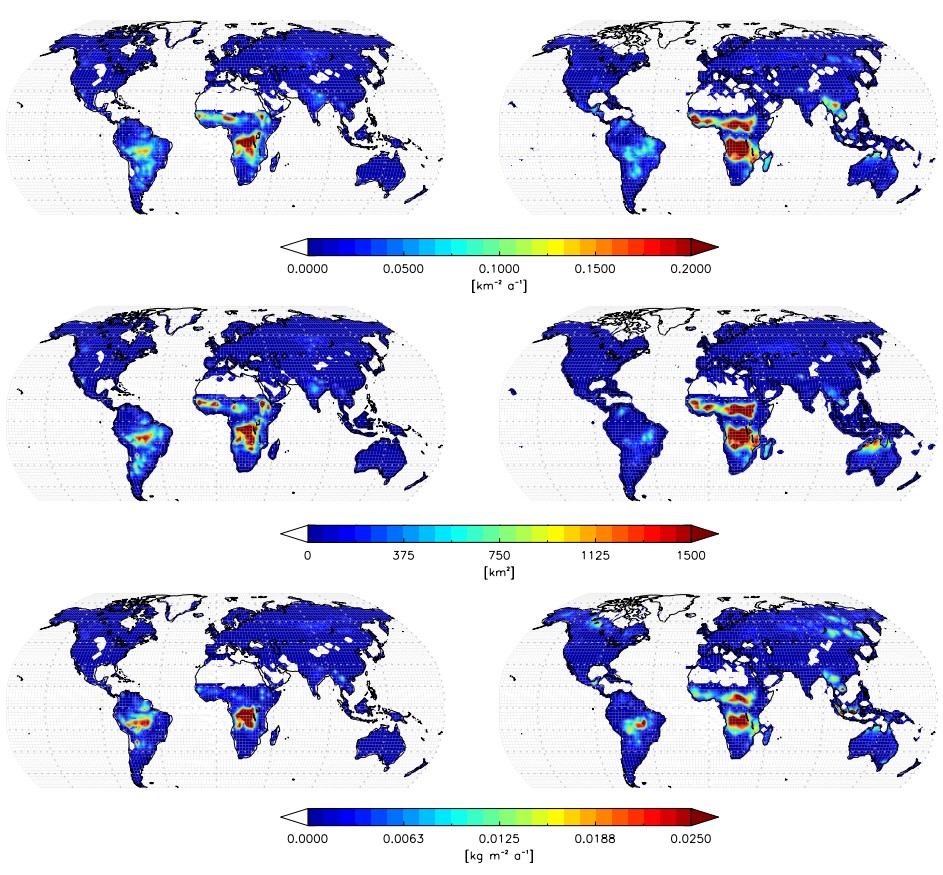

1300

Figure 8: Annual mean model (left) and satellite based (right) fire count (upper), burned

area (middle), and CO emissions (lower). Modeled annual mean is based on an ensemble

of 10 simulations. Satellite detected fire count are based on MODIS Aqua retrievals of

2003-2016, burned area is based on GFED4s inventory of 2003-2016, and CO emissions

are based on climatological GFED4s emissions of 1995-2010.



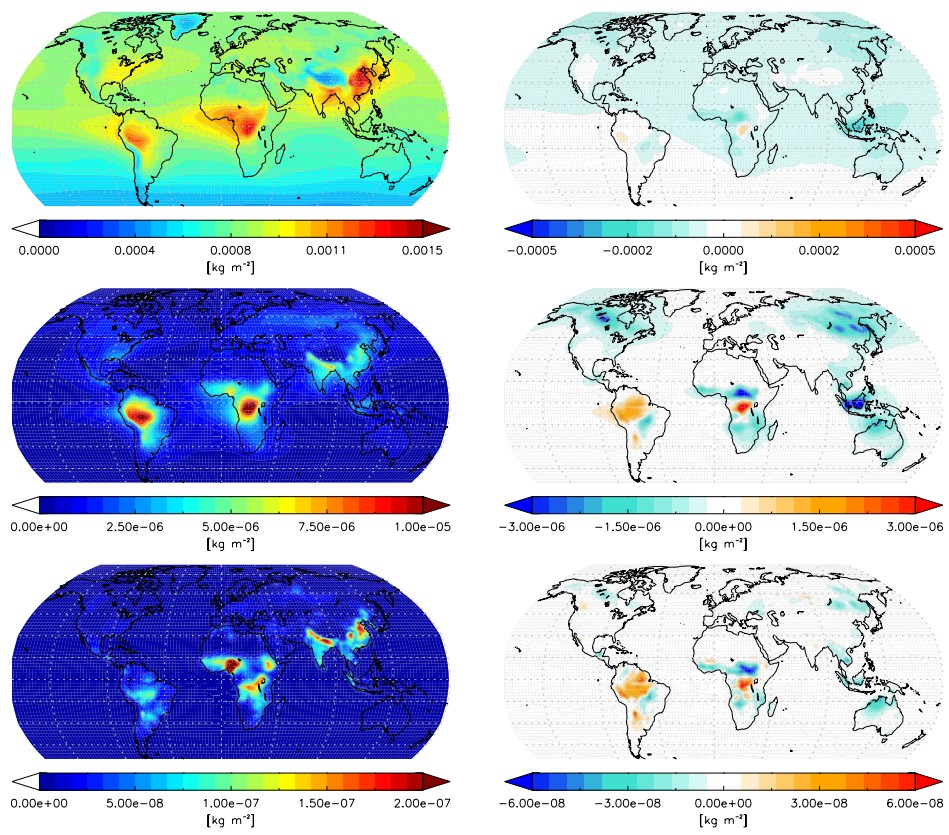


Figure 9: Modeled annual mean column density using pyrE fire emissions (left), and the
difference in column densities with a simulation using offline GFED4s emissions (pyrE −
GFED4s; right). CO (upper), OA (middle), and BC (lower). Data based on an ensemble
of 10 simulations.


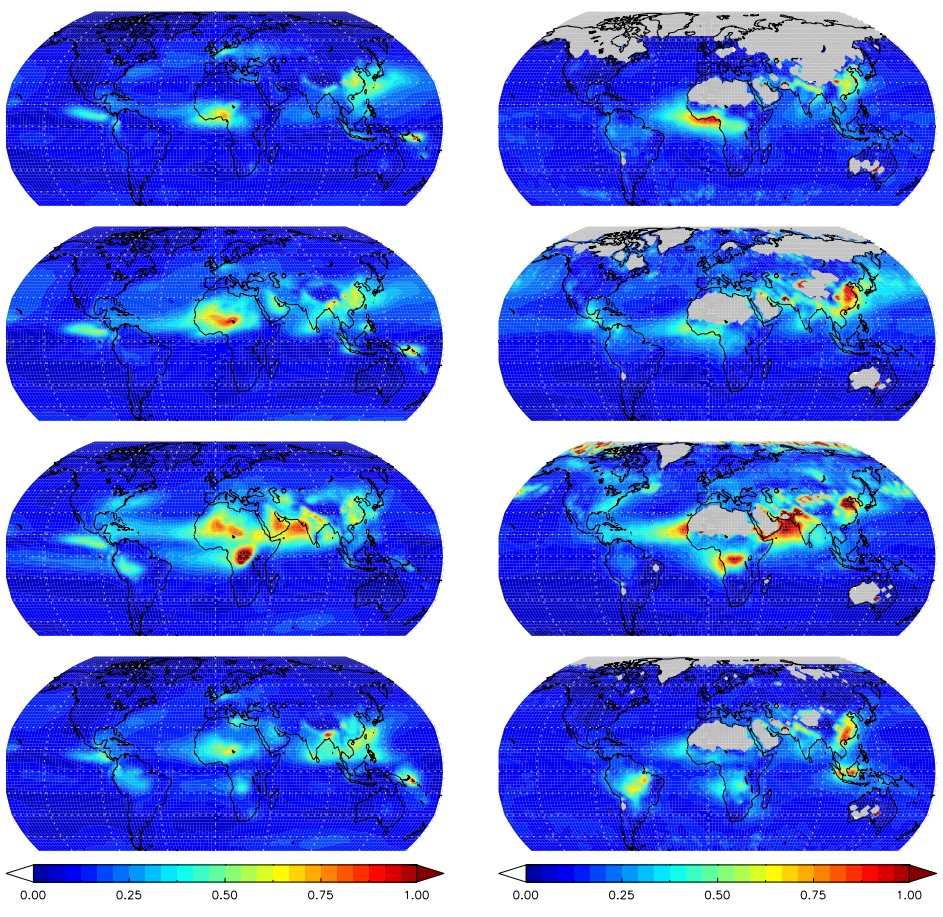


Figure 10: Monthly modeled clear-sky aerosol optical depth (AOD) simulated using pyrE
fire emissions (left), and detected by Aqua-MODIS (right). January (first row), April
(second row), July (third row), and October (last row). Monthly mean simulated AOD is
based on an ensemble of 10 simulations, and climatologically monthly MODIS AOD is
based on 2003-2007 data. Missing MODIS data is shaded in light gray.







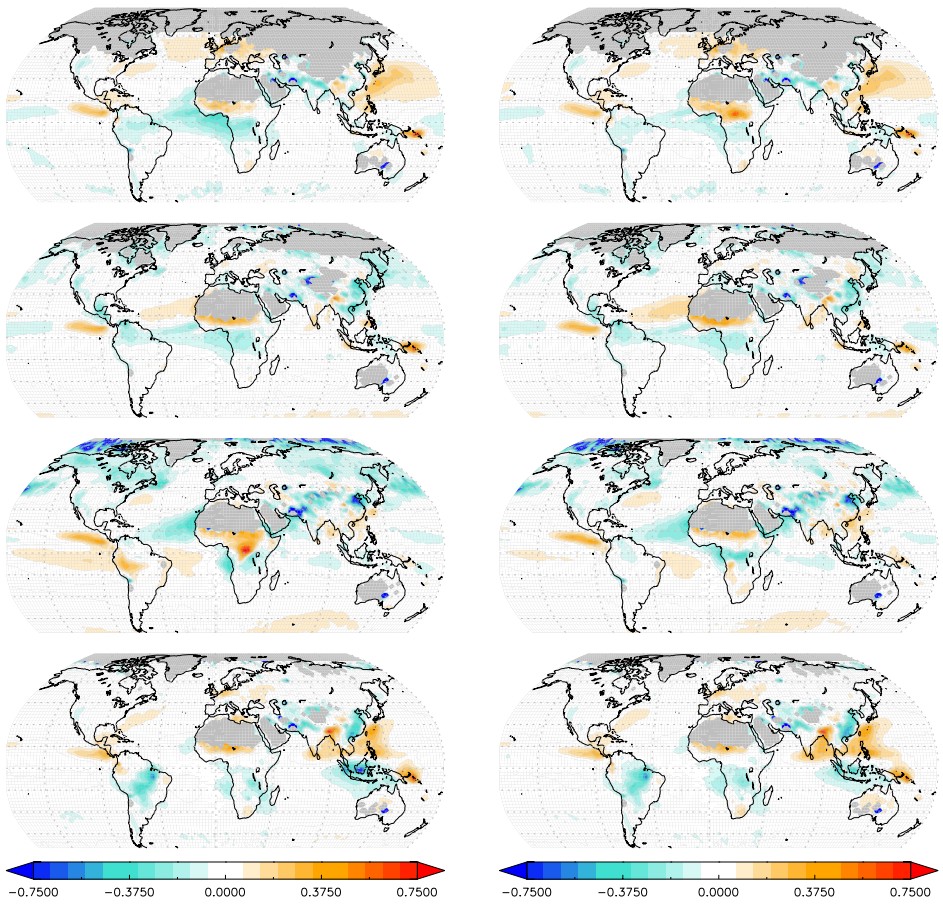

Figure 11: The difference in monthly modeled clear-sky aerosol optical depth (AOD) and MODIS Aqua (model – satellite). Model simulations using pyrE fire emissions (left) and model simulations using offline GFED4s emissions (right). January (first row), April (second row), July (third row), and October (last row). The difference is based on an ensemble of 10 simulations and 2003-2007 MODIS climatological monthly data. Missing MODIS data is shaded in light gray.



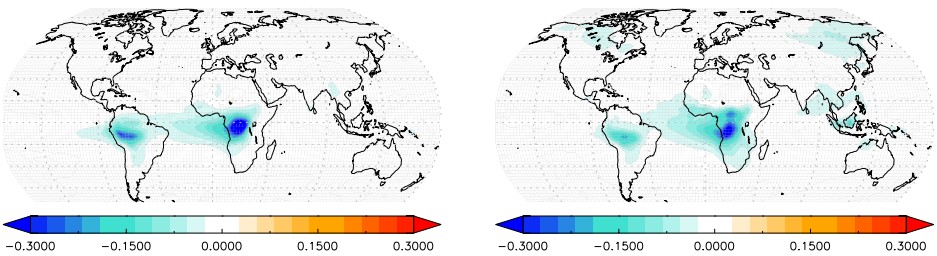


Figure 12: The difference in annual modeled clear-sky aerosol optical depth (AOD)
between a simulation with no fire emissions to a simulation using pyrE fire emissions
(left), and a simulation with offline GFED4s emissions (right). The difference (model
with no fire emissions – model with fire emissions) is based on an ensemble of 10
simulations.


















## APPENDIX


Figure A1: Seasonality of total fire count (FC) detected by MODIS Aqua (red) and Terra
(orange) and simulated (blue) in all GFED regions (Fig. 1). Error bars represent the 10-
year range in the simulations. Note the different scale in each panel.









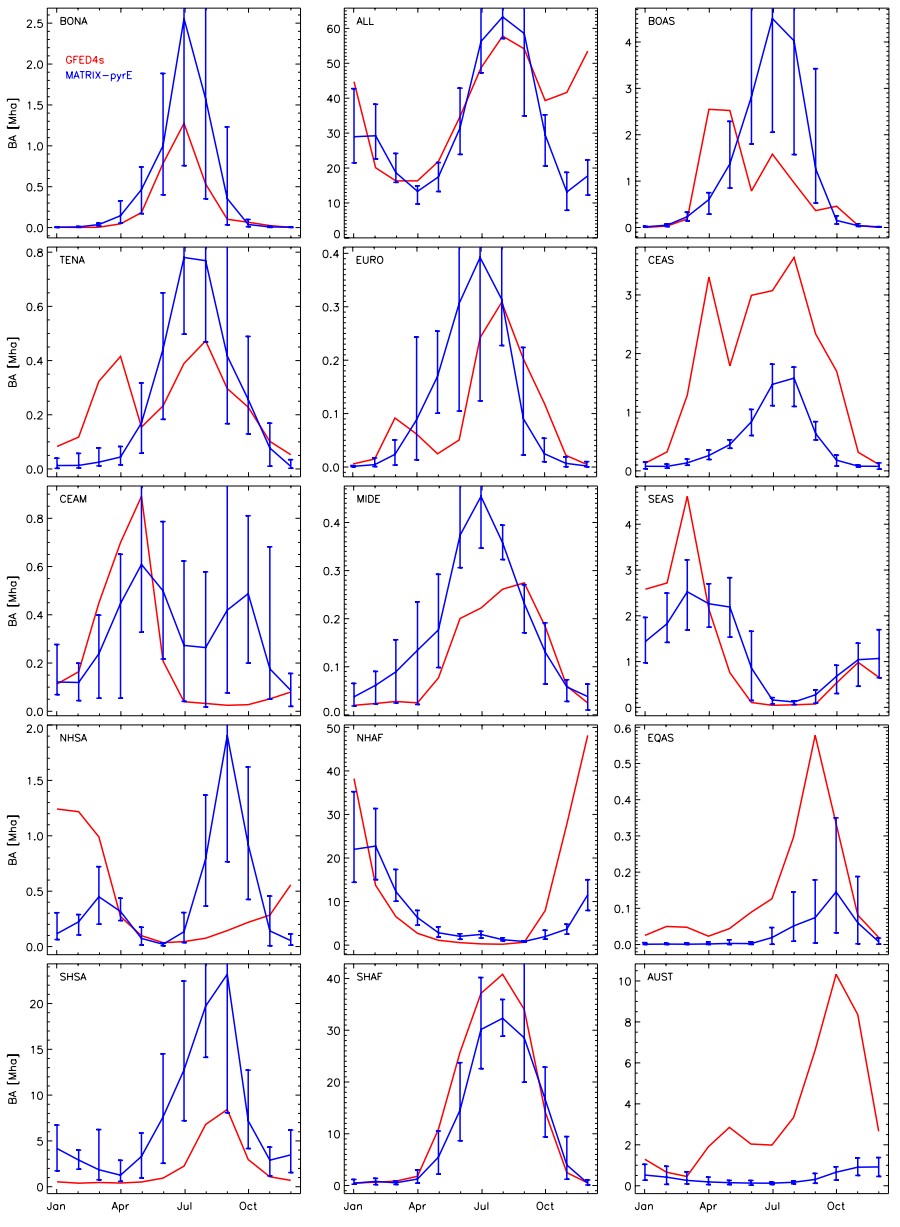


Figure A2: Seasonality of total burned area; simulated (blue) and reported by GFED4s
(red) in GFED regions. Error bars represent the 10-year range in the simulations. Note the
different scale in each panel.






Figure A3: Seasonality of total fire CO emissions; simulated (blue) and reported by
GFED4s (red) in GFED regions. Error bars represent the 10-year range in the simulations.
Note the different scale in each panel.






Figure A4: Seasonality of total fire organic aerosol (OA) emissions; simulated (blue) and
reported by GFED4s (red) in all GFED regions. Error bars represent the 10-year range in
the simulations. Note the different scale in each panel.






Figure A5: Seasonality of total fire BC emissions; simulated (blue) and reported by
GFED4s (red) in all GFED regions. Error bars represent the 10-year range in the
simulations. Note the different scale in each panel.