# Peer review of "The interactive global fire module pyrE"

_Geoscientific Model Development, 2019_

## Referee Comment (RC1) · Anonymous Referee #1 · 11 Nov 2019

General comments In this manuscript, the authors describe pyrE, a new fire module for the ModelE Earth System Model. pyrE builds upon work by previous authors but includes some novel elements that could be of interest to the broader community of global fire modelers. The authors present not only the direct outputs of pyrE (which performs acceptably, especially considering the huge variation in performance seen in global fire models), but also evaluate its impact on ModelE's representation of atmospheric chemistry, specifically with regard to aerosol optical depth. There is nothing especially groundbreaking presented here, but the manuscript represents a well-written and (mostly) thorough documentation of an important part of an Earth system model—something exactly appropriate for publication in this journal. However, the authors need to be much clearer about the choices they had to make because of limitations of their vegetation model, and better place these choices into the context of previously published fire models.

[Figure]

My main criticism has to do with the authors' contextualization of their decision to tie emissions to fire count. This begins in the Abstract:

"Fire emissions are generated from the actual flaming phase in pyrE (fire count), not the scar left behind (burned area), as is commonly done in other interactive fire modules."

Continues at the end of the Introduction:

"pyrE uses fire count to derive emissions, and is therefore more directly connected to the actual fires, in contrast to other fire models that use BA, a measure more indicative of fire's effect on the landscape."

And shows up again at the beginning of Section 5.3:

"Due to the intricate processes involved in burned area spread, most fire models struggle to reproduce the observed trend [Andela et al., 2017] and seasonality [Hantson et al., 2017a] of burned area. A more direct approach would be to use fire count, similar to the approach of Pechony and Shindell (2009, 2010) and Pechony et al. (2013)."

In the real world, fire emissions are a product of (a) how much area burns, and (b) how much fuel is combusted per square meter of burned area. Burned area is not just some side product that vegetation models need to look at to count how many trees die; it's the only way to know exactly how much fuel could possibly be combusted in a fire, and fuel combustion is what creates emissions. Thus, it is flatly incorrect to assert that bypassing burned area makes pyrE "more directly connected to the actual fires." The burned area IS the "actual fire."

It is unclear from this manuscript whether Ent, the part of ModelE that represents vegetation, actually simulates biomass in any meaningful way. Presumably it does not, which makes the use of fire count-based emissions factors acceptable—modelers must sometimes do what is possible given existing structures. (If it DOES simulate biomass, I would say that pyrE would need to be completely reworked and this manuscript rejected.) However, THAT is how this decision should be framed—as the best that can be

done given an extremely basic representation of the terrestrial biosphere. The authors should not attempt to position their method as something superior to what is done in nearly every other global fire model, which are integrated with seemingly superior vegetation models. (It's possible I'm reading too much into what the authors have written, and that they're not actually trying to position it that way, but my overall point remains: The authors need to explain that what they are doing is a kludge to work around a deficit in their vegetation model.)

This is especially galling considering the authors' complete omission of SPITFIRE-based fire models (three presented in Rabin et al., 2017 and supplement, which describes the models participating in the Fire Model Intercomparison Project [FireMIP]) from the Discussion. These and other models not only calculate process-based fire counts and burned area, but also process-based fireline intensity and fuel consumption. THOSE models are "directly connected to the actual fires." But no, the only text hinting at those models' capabilities is the second-to-last sentence of the Conclusions: "Almost no fire models include fire energy." Three of the eleven fire models presented in the FireMIP protocol paper is hardly "almost no models"!

It seems that the authors have chosen fire counts rather than burned area because, as they point out in the Results, pyrE performs better for fire count than for burned area. But did the authors ever actually test whether parameterizing emissions factors based on fire counts actually gives better results than parameterizing based on burned area? This should be tested and presented, at least in a Supplement. In general, the authors need to present much more information about their parameterization methods and results. All we see about the parameterization of emissions factors is the input data and a reference to multivariate curve fitting—much more information is needed in the interest of reproducibility.

Dancing around this deficiency in their vegetation model, rather than addressing it head-on, seems to pop up in other parts of the manuscript. For example, the authors do not really present any evidence for the idea that anthropogenic fire suppression in

the Middle East is as strong as it is in the United States. This does not really make sense, given that, as the authors point out, there is so little biomass across much of the Middle East. Instead, the issue is more likely that pyrE does not consider fuel availability at all in calculating fire counts. Again, a kludge to deal with this is something that is justifiable, and the authors need to address it directly. This also touches on the need for more transparency about parameterization. The authors need to present literature evidence for the elimination of suppression in Africa, as well as for the new parameterization for the US. The first paragraph of Section 2.3 relies heavily on broad statements that are not backed up by any citations.

This is unrelated but a potentially large issue: The authors are not interested in cropland fires, which makes sense given their model system's limitations, but why then do they not filter out MODIS hotspots and GFED emissions based on which were detected on cropland? One of the MODIS products is a land cover map (three, actually) that has been used in previous work to filter out hotspots detected on cropland, either for the purpose of discarding them or analyzing them on their own.

It's also worth pointing out that the authors commit an all-too-common mistake in conflating MODIS hotspot detections with "fire counts." One large fire might have a fireline long enough for multiple hot pixels to be detected; a slow-moving fire might result in pixels that are counted multiple times as hotspots despite being part of one fire. The authors' use of the hotspot data itself is not necessarily flawed—it's fine for the "fire counts" parameterization to target hotspots since it's all ideally going to get worked out in the emissions factors—but the authors should revise the manuscript to clarify exactly what it is the remote sensing data show. A true "fire counts" product is something more like the Global Fire Atlas (https://www.earth-syst-sci-data.net/11/529/2019/).

Minor comments

- Some parts of the manuscript are too detailed and/or technical. For example, the authors spend over a page in the Introduction discussing the ways that people use fire

to manage land, and exactly when those land management fires occur. For a process that's not even represented in the model, that seems like a pretty big waste of space. I would prefer to see relevant information along these lines presented instead in the Results and/or Discussion, to provide context for poor performance in some regions. (The authors do a bit of this, but more would be an improvement.) It is also unnecessary to get into the technical details of remote sensing, such as the sensor channels used to detect fire counts or the reflectance characteristics used in calculating burned area.

- It seems that burned area is only used as an input to flammability in subsequent time steps. This should probably be made clearer, given that burned area is a primary product of most existing fire models. More importantly, though: How long does it take for that effect to fade? Is it just how much of the grid cell has burned EVER? Surely not, but the authors don't specify how this works.

- Why do the white (zero) areas on the right (model-simulated) side of Fig. 8 not match up exactly?

- Figures throughout could use more labeling. It makes it hard to interpret figures when the reader needs to keep going back to the caption to figure out what's in the right vs. left column, top vs. bottom row, etc.

- I understand what the authors are getting at here given the context (at the end of the presentation of results of the simulation at times of day equivalent to the MODIS overpasses) but this sentence makes little sense and should be reworked. It sounds like "Even though A=B, A>B.": "The implications of these findings are that even though the simulated monthly mean fire count is in the range of Terra and Aqua (Fig. 4, A1), the simulated fire count is in fact higher than MODIS retrievals." [Lines 541–543]

- These related sentences were extremely confusing until eventually I remembered about how previously-burned area affects flammability; this should be clarified: - "Nevertheless, even with this large correction factor, burned area has a very minor impact on fire count and fire emissions as it accounts for a small fraction of the grid cell that is

able to burn." (lines 382–384) - "burned area itself has a minor impact on fires due to its small percentage in a grid cell" (lines 577–578)

- I first noticed this at Line 588 and following, but it may have occurred earlier: The authors seem to sometimes incorrectly refer to the NHAF region as "sub-Saharan Africa." Sub-Saharan Africa in fact refers to ALL of Africa south of the Sahara, not just the northern-hemisphere portion.

Technical corrections

- "Bias-high" and "bias-low" throughout should be "biased high" and "biased low".

- Line 618: Should be cold- and drought-deciduous

- Line 663: Extra period

- Line 670: Missing word

- Line 677: Incorrect capitalization of pyrE

- Line 684: Extra comma

- Line 699: Missing comma at end of line

- Line 700: Should be Middle EAST, presumably

- Fig. A1 BONA: Truncated "1" at top of Y axis

―――――――――――――――――――

---

## Short Comment (SC1) · 15 Nov 2019

Dear authors,

in my role as Executive editor of GMD, I would like to bring to your attention our Editorial version 1.2:

https://www.geosci-model-dev.net/12/2215/2019/

This highlights some requirements of papers published in GMD, which is also available on the GMD website in the 'Manuscript Types' section:

http://www.geoscientific-model-development.net/submission/manuscript_types.html

In particular, please note that for your paper, the following requirements have not been met in the Discussions paper:

- The main paper must give the model name and version number (or other unique identifier) in the title.

- "Code must be published on a persistent public archive with a unique identifier for the exact model version described in the paper or uploaded to the supplement, unless this is impossible for reasons beyond the control of authors. All papers must include a section, at the end of the paper, entitled "Code availability". Here, either instructions for obtaining the code, or the reasons why the code is not available should be clearly stated. It is preferred for the code to be uploaded as a supplement or to be made available at a data repository with an associated DOI (digital object identifier) for the exact model version described in the paper. Alternatively, for established models, there may be an existing means of accessing the code through a particular system. In this case, there must exist a means of permanently accessing the precise model version described in the paper. In some cases, authors may prefer to put models on their own website, or to act as a point of contact for obtaining the code. Given the impermanence of websites and email addresses, this is not encouraged, and authors should consider improving the availability with a more permanent arrangement. Making code available through personal websites or via email contact to the authors is not sufficient. After the paper is accepted the model archive should be updated to include a link to the GMD paper."

Therefore a version number of pyrE must be added in the title of the revised manuscript. Additionally, the exact code version on which this publication is based should be archived in a permanent archive providing a permanent identifier (e.g. DOI).

Yours,

Astrid Kerkweg

---

## Referee Comment (RC2) · Anonymous Referee #2 · 15 Jan 2020

Review for Mezuman et al., The interactive global fire module pryE

General comments: The present paper presents a new fire module as implemented in the GISS ModelE ESM. The fire module relies on previous fire model developments. It is nice to see new developments in fire modeling, here focusing on fire emissions. Overall the manuscript is well written and the results presented clearly. PyrE does not seem to outperform existing fire models (which was clearly not the objective either), but its performance is neither bad compared to the performance of other fire models. My main issue with the current manuscript is that it is assuming fire numbers and active fire counts to be the same thing, which is confusing and can lead to wrong interpretation of results. While I don't have any problem with the chosen approach of moving directly from fire count to emissions, the authors should present their methods in a more coherent way. Below are some comments which I hope will explain in more detail where I think the issues are.

[Figure]

Detailed comments:

L35: pyrE: Is this an acronym? If so, indicate this.

L40: "...as is commonly done in other interactive fire modules. ", this is not true.

L98-123: It is unclear to me why you have here such a huge emphasis on regional seasonality of fires. It distracts a bit from the main message you want to bring here, which is presented in the following section where you do explain the reasons why you are so much interested in incorporating a fire module within your ESM.

L216: "fire counts" are not really a much-used term within the fire community, thus making it hard to understand what you mean. Use "number of fires" or similar to avoid confusion. Fire counts could be interpreted as thermal anomaly detections from satellites etc., which is something completely different.

L220-223: while I think I know what you want to indicate here (flaming fase compared to area burnt), I don't agree with this. While this could be a computationally faster way (which can be an argument), there are also inconveniences associated, with e.g. no fuel gradient driving your emission amounts within each region. Again, I don't have anything against your approach, but you should explain things clearer to avoid confusion.

L370-373: may add this to section 4 and change that name to "datasets", as it has nothing to do with ModelE.

L372: "and on future projections (not used in this study) for years past 2010 ". If it is not applicable here, no need to mention it.

L375-379: This is not really accurate, as some models here just assign a fixed fire size instead of a dynamic one, others are completely empirical and hence don't simulate fire numbers etc.

L379-380: This is confusing: during the methods you indicate that you produce number

of fires and that you incorporate a fire growth method to obtain burnt area. Hence your fire counts = fire numbers. However, here it seems to indicate that the authors of the manuscript use two completely different concepts "fire number" and "satellite observed thermal anomalies" indiscriminate while they cannot be compared one-to-one. Thermal anomalies/active fire counts are detections of whether within a pixel at the time of the satellite overpass there is an fire actively burning (or smouldering), so you can have 100s or 1000s of active fire detections within 1 single fire (and hence number of thermal anomalies is closely related to burnt area and not number of fires at a regional scale). So, one would expect to have more active fires than number of fires, and hence the need of a scaling factor. This is a problem within this manuscript and should be fixed. As you need an even much larger scaling factor for burned area it mainly seems that your fires don't grow large enough? As a side note, and again, no problem if you ignore this. Have you thought about ignoring burnt area completely, as it barely used (no terrestrial impact as far as described in the manuscript) and doesn't seem to influence your results much?

L403: why 2005 and not a climatology as well ? Now you focus on a specific year which its peculiarities (El Niño with much fire in the amazon).

L409: This is personal, but I find this section way to detailed for just describing some of the reference datasets used. Especially considering the brief introduction to modelE, which I think is more important (at which time resolution do you run?), Especially the integration of the fire module within the land surface model. Does the fire model change albedo? Carbon stocks? LAI? I don't think these important parts are covered, and would prefer that compared to these detailed descriptions of external datasets.

L471: GFED4s starts in 1997 (1995 is mentioned a couple of times more in the manuscript)

L522: This is great and the first time I see (output) of sub-daily fire model results. I think it would be nice that it is indicated in the methods as well. So, this is completely driven

by changes in VPD, which seems indeed to capture the expected sub-daily dynamics surprisingly well. Have you thought of comparing this to GFED 3h estimated emissions or thermal anomalies from some geostationary satellite? Just a thought, so don't feel obliged.

L554-555: I guess you divided these numbers by accident by 10. GFED4s burnt area is somewhere around 450Mha (did you use GFED4 instead which is around 350Mha?)

Table 2: Indicate time period.

Figures: increase fond size of most figures, as many legends and axis are now barely readable.

Figure 3 can go to supplementary, as burnt area is anyway not the main focus of the fire module.

Figure 5: why didn't you indicate the values for Terra and Aqua instead of a star. If you don't want to show the values, I think the stars are a bit misleading as they are put straight on the mean model output, so you better draw a shaded box, line or something similar.

Figure 7-9: indicate which map indicates what, be it with letters or a description above each subplot.

---

## Referee Comment (RC3) · Anonymous Referee #3 · 19 Jan 2020

This paper describes the updates of pyrE, the fire component of GISS ModelE. The discussion of the paper is written very well, demonstrating the comprehensive grasp of fire modeling by the authors. If this were the first paper describing pyrE, I would firmly support its publication. However, 10 years after the papers by Pechony and Shindell, the current updates of pyrE do not appear to have sufficient improvements to justify its publication. My guess is that the updates of pyrE proved to be more difficult than the researchers anticipated. As an important component of the venerable ModelE, pryE updates obviously need to be documented. I wonder if it would be better to include pryE updates as part of another ModelE paper rather than a standalone paper.

In model description, sections 2.1, 2.2, 2.4, and probably some of 2.5 are essentially the same as Pechony and Shindell (2009). For a journal paper, these sections should be summarized in 1-2 paragraphs and the authors can refer readers to Pechony and Shindell (2009).

[Figure]

Section 2.5 is new. However, burned area (BA) modeling is odd in this paper. The purpose of developing it is unclear. The introduction section is 7 pages long, but it did not include discussion on why modelE needs to simulate BA when fire emissions are calculated using fire counts. This is not a review paper. The introduction section is too verbose and should be shortened substantially. The BA modeling is based heavily on Li et al. (2012) and the biases compared to GFED4s are quite large (Figure 8). It seems to be worse than the other global fire models cited in the paper. If BA modeling does not serve a useful purpose for modelE, it should be removed.

I am alarmed by section 2.7. Line 368 states lightning ignition is scaled down by a factor of 10, but line 380 states that fire counts are scaled up by a factor of 30. These statements do not inspire confidence in modelE simulations. pyrE is supposed to be a physical model. Does this large a scaling factor imply that the model is not really physical? How is 30 determined? With such a large arbitrary scaling factor as a tuning knob, why is the scaling factor not tuned such that the simulated global fire counts agree with satellite observations (rather than a 32-42% low bias)? One can go a step further to tune this factor differently for each region. For modelE, it seems that a statistical fire model would work better than the current setup.

Line 381 states that BA is scaled up by a factor of 250. After discounting the factor of 30 in fire count scaling, BA is still scaled up a factor 8. Li et al. (2012) did not have to do this large scaling in their model. What went wrong here?

Sections 2.3 and 5.1.1 are new improvements to pyrE. Discuss how Eq. (6) was derived. Is it through some kind of linear regression?

The evaluation in section 2.6 is inadequate. Fire emissions are what matter to modelE simulations. Regional emission biases of pyrE OA and BC relative to GFED4s should be discussed. The column load comparison in Table 2 is not informative and should be removed. If fire emission is underestimated but the column load is not, it only shows that fire emissions are unimportant (and pyrE is not needed). Fire emissions may not

significantly change global mean column loads but they strongly affect the atmosphere, which the authors described in the introduction.

In the abstract, line 40-42 states "Using pyrE, we examine fire behavior, regional fire suppression, burned area, fire emissions, and how it all affects atmospheric composition." In the current setup, fire behavior and burned area do not affect atmospheric composition in modelE.

In the abstract, the last sentence, "Yet, in terms of AOD, a simulation with interactive fire emissions performs just as well as simulation with prescribed fire emissions", is misleading. It sounds as if global fire emission biases as large as 40% do not matter for modelE simulations, which I think is not what the authors meant. The pyrE and GFED4s simulations of AOD are very different in regions where fire emissions are present and large.

Figure 12 shows that fire AOD effect is only about 10% (line 676-677). The AOD evaluation in this paper should be about fire emissions not modelE AOD simulation. Figures 9 should compare fire-only AOD between pyrE and GFED4s simulations. The differences are large. Figure 11 is not useful because the non-fire AOD model biases are much more apparent than fire AOD.

---

## Author Comment (AC1) · 4 May 2020

We would like to thank the referees for their comments and in depth review of this paper. We have addressed all comments as described below, where the referee comments appear in bold, our responses are in italic below. An important point to make is that we renamed "fire count" to "active fires" throughout the manuscript.

We would like to add that this manuscripts' main goal is to describe the fire model pyrE, that is now an interactive component of modelE, the NASA GISS Earth System Model. The fire module is based on an earlier off-line code development by Pechony and Shindell (2009,2010), and has been extended in its functionality, which is explained in detail in the paper. Now that the fire model is an integral part of the climate model, a detailed description and evaluation as presented here seemed essential from our point of view, in order to provide a full description of the functionalities of modelE.

[Figure]

Response to Referee #1: General comments In this manuscript, the authors describe pyrE, a new fire module for the ModelE Earth System Model. pyrE builds upon work by previous authors but includes some novel elements that could be of interest to the broader community of global fire modelers. The authors present not only the direct outputs of pyrE (which performs acceptably, especially considering the huge variation in performance seen in global fire models), but also evaluate its impact on ModelE's representation of atmospheric chemistry, specifically with regard to aerosol optical depth. There is nothing especially groundbreaking presented here, but the manuscript represents a well-written and (mostly) thorough documentation of an important part of an Earth system model— something exactly appropriate for publication in this journal. However, the authors need to be much clearer about the choices they had to make because of limitations of their vegetation model, and better place these choices into the context of previously published fire models. My main criticism has to do with the authors' contextualization of their decision to tie emissions to fire count. This begins in the Abstract:

---

## Author Comment (AC2) · 4 May 2020

We would like to thank the referees for their comments and in depth review of this paper. We have addressed all comments as described below, where the referee comments appear in bold, our responses are in italic below. An important point to make is that we renamed "fire count" to "active fires" throughout the manuscript.

We would like to add that this manuscripts' main goal is to describe the fire model pyrE, that is now an interactive component of modelE, the NASA GISS Earth System Model. The fire module is based on an earlier off-line code development by Pechony and Shindell (2009,2010), and has been extended in its functionality, which is explained in detail in the paper. Now that the fire model is an integral part of the climate model, a detailed description and evaluation as presented here seemed essential from our point of view, in order to provide a full description of the functionalities of modelE.

**Response to Referee #1:**

General comments In this manuscript, the authors describe pyrE, a new fire module for the ModelE Earth System Model. pyrE builds upon work by previous authors but includes some novel elements that could be of interest to the broader community of global fire modelers. The authors present not only the direct outputs of pyrE (which performs acceptably, especially considering the huge variation in performance seen in global fire models), but also evaluate its impact on ModelE's representation of atmospheric chemistry, specifically with regard to aerosol optical depth. There is nothing especially groundbreaking presented here, but the manuscript represents a well-written and (mostly) thorough documentation of an important part of an Earth system model something exactly appropriate for publication in this journal. However, the authors need to be much clearer about the choices they had to make because of limitations of their vegetation model, and better place these choices into the context of previously published fire models.

My main criticism has to do with the authors' contextualization of their decision to tie emissions to fire count. This begins in the Abstract:

1. "Fire emissions are generated from the actual flaming phase in pyrE (fire count), not the scar left behind (burned area), as is commonly done in other interactive fire modules."

*Rephrased the sentence to: "Fire emissions are generated from the flaming phase in pyrE (active fires)"*

2. Continues at the end of the Introduction:

"pyrE uses fire count to derive emissions, and is therefore more directly connected to the actual fires, in contrast to other fire models that use BA, a measure more indicative of fire's effect on the landscape."

Edited the sentence to only highlight the difference with other fire models: "pyrE uses active fires to derive emissions in contrast to other fire models that use BA"

3. And shows up again at the beginning of Section 5.3:

"Due to the intricate processes involved in burned area spread, most fire models struggle to reproduce the observed trend [Andela et al., 2017] and seasonality [Hantson et al., 2017a] of burned area. A more direct approach would be to use fire count, similar to the approach of Pechony and Shindell (2009, 2010) and Pechony et al. (2013).

We deleted the beginning of this sentence and rephrased the rest: "Due to limitations in the current capabilities of the simulated terrestrial biosphere in ModelE emissions are generated from active fires, similar to the approach of Pechony and Shindell (2009, 2010) and Pechony et al. (2013)."

In the real world, fire emissions are a product of (a) how much area burns, and (b) how much fuel is combusted per square meter of burned area. Burned area is not just some side product that vegetation models need to look at to count how many trees die; it's the only way to know exactly how much fuel could possibly be combusted in a fire, and fuel combustion is what creates emissions. Thus, it is flatly incorrect to assert that bypassing burned area makes pyrE "more directly connected to the actual fires." The burned area IS the "actual fire."

We agree with the Referee that the fuel combusted is key in understanding emissions, but we can't get that either from active fires or burned area, without using a parameterization or a scaling factor that is linked with the underlying vegetation that was impacted by the fire. We disagree that burned area is the actual fire; the actual fire is the active fire, which is the new phrasing we use instead of fire count (see earlier replies),

**while burned area is what is left behind after the active fire moved elsewhere or got extinguished.**

It is unclear from this manuscript whether Ent, the part of ModelE that represents vegetation, actually simulates biomass in any meaningful way. Presumably it does not, which makes the use of fire count-based emissions factors acceptable modelers must sometimes do what is possible given existing structures. (If it DOES simulate biomass, I would say that pyrE would need to be completely reworked and this manuscript re- jected.) However, THAT is how this decision should be framed—as the best that can be done given an extremely basic representation of the terrestrial biosphere. The authors should not attempt to position their method as something superior to what is done in nearly every other global fire model, which are integrated with seemingly superior vegetation models. (It's possible I'm reading too much into what the authors have written, and that they're not actually trying to position it that way, but my overall point remains: The authors need to explain that what they are doing is a kludge to work around a deficit in their vegetation model.)

As already stated in the manuscript, land cover is given as input to Ent and ModelE and it is not being modified dynamically by Ent. The only interaction of Ent with pyrE is by providing LAI as stated in L361, L389-L393, L608-609 (submitted version). Additionally, we added the following sentence in L328-331 (revised version): "The use of active fires to derive emissions is driven by the extremely rudimentary representation of the terrestrial biosphere in ModelE, under which interactive fuel consumption cannot be calculated."

4. This is especially galling considering the authors' complete omission of SPITFIRE- based fire models (three presented in Rabin et al., 2017 and supplement, which describes the models participating in the Fire Model Intercomparison Project [FireMIP]) from the Discussion.

We have cited SPITFIRE relevant work, specifically JSBACH-SPITFIRE in L64 (Hantson et al., 2015), L190 (Lasslop et al., 2014) as well as the Firemip papers Hantson et al., 2016, Hantson 2017a and Rabin et al., 2017 in L190, L205, L378, L582 (submitted version). See the next reply for additions in the text.

5. These and other models not only calculate process-based fire counts and burned area, but also process-based fireline intensity and fuel consumption. THOSE models are "directly connected to the actual fires." But no, the only text hinting at those models' capabilities is the second-to-last sentence of the Conclusions: "Almost no fire models include fire energy." Three of the eleven fire models presented in the FireMIP protocol paper is hardly "almost no models"!

We added in L176-L180 (revised version): "The most sophisticated models are coupled to dynamic global vegetation models and directly connect fire-Earth system interactions through fuel consumption (e.g. LPJ-GUESS-GlobFIRM and LPJ-GUESS-SIMFIRE-BLAZE (Smith et al., 2001, 2014; Lindeskog et al., 2013), and MC-Fire (Bachelet et al., 2015; Sheehan et al., 2015))."

Also, we rephrased the sentence in L746-L749 (revised version):

*"Finally, given that the heat component of fires interact with the climate system, and can also be used to derive more accurate emissions, as demonstrated by Ichoku and Ellison (2014) and three of the eleven FireMIP models (Rabin et al., 2017) , it is worthwhile taking it into consideration when developing new fire modeling capabilities."*

6. It seems that the authors have chosen fire counts rather than burned area because, as they point out in the Results, pyrE performs better for fire count than for burned area. But did the authors ever actually test whether parameterizing emissions factors based on fire counts actually gives better results than parameterizing based on burned area? This should be tested and presented, at least in a Supplement.

We chose to use fire count (active fires in the revised manuscript) because it is the active phase of the fire and developed parameterization for this. The active phase of the fire is the time at which most of the emissions are happening. Also note support from Referee #2: "I don't have any problem with the chosen approach of moving directly from fire count to emissions". Developing an additional parameterization for burned area, just for the sake of comparing the two approaches will not improve our parameterization. We added in L150-151 (revised version):

"Globally, most fire emissions occur during the active phase of the fire, with peat fires as the main exception [Andreae, 2019]."

To clarify this we also edited L326-331 (revised version):

"Trace gas and aerosol emissions are generated during the active phase of the fire and are calculated as the product of simulated active fires and emission factors and are a function of PFT (denoted by v) and chemical specie (denoted by s). The use of active fires to derive emissions is driven by the extremely rudimentary representation of the *terrestrial biosphere in ModelE, under which interactive fuel consumption cannot be calculated. "*

7. In general, the authors need to present much more information about their parameterization methods and results. All we see about the parameterization of emissions factors is the input data and a reference to multivariate curve fitting much more information is needed in the interest of reproducibility.

We edited L340-346 (revised version) and expanded to include the following: "Our technique, known as multivariate curve fitting, matched the emissions within the PFT fraction of the grid cell with the respective active fires. We correlated a time series of GFED4s emissions with a time series of MODIS fire count for each modeled PFT in a grid cell. Our settings included statistical (Poisson) weighting of the GFED4s emissions (1 over emissions) and a uniform initial estimate of 100,000 kg m-2 s-1 per fire per PFT. This calculation resulted in a specific emission factor per PFT (Table 1). "

8. Dancing around this deficiency in their vegetation model, rather than addressing it head-on, seems to pop up in other parts of the manuscript. For example, the authors do not really present any evidence for the idea that anthropogenic fire suppression in the Middle East is as strong as it is in the United States. This does not really make sense, given that, as the authors point out, there is so little biomass across much of the Middle East. Instead, the issue is more likely that pyrE does not consider fuel availability at all in calculating fire counts. Again, a kludge to deal with this is something that is justifiable, and the authors need to address it directly. This also touches on the need for more transparency about parameterization. The authors need to present literature evidence for the elimination of suppression in Africa, as well as for the new parameterization for the US. The first paragraph of Section 2.3 relies heavily on broad statements that are not backed up by any citations.

The fire model is independent of Ent, it uses some of its output. The change in fire suppression was only made as a method to make the model match measurements better. We do not claim that this is based on sophisticated parameterization that is applied to the underlying vegetation. See L197-199 (submitted version): "Some models also include simplified empirical relationships of anthropogenic ignition and suppression, which, at present, are not understood in a dynamic process level". The factors we used helped improve model results in those regions and their exact values (including factors in other regions) need to be addressed in future research.

We could not find an indication in the literature that active fire suppression is exercised at large scales in Africa. Our assumption of flammability and ignition controlled fire regime in Africa is supported by Archibald (2016) that describes that in Africa people indirectly suppress fires by impacts on fuel amount and fuel continuity. Parisien and Mortiz (2009) and Marlon et al., 2012 support our assumptions regarding intense active fire suppression in the USA. We could not find adequate literature about fire suppression in the MIDE, but since it is a very small area that does not affect in any significant way the global fire behavior, we decided to keep the modified suppression parameter that provides a better agreement with GFED4s.

We edited L268-L275 (revised version): "For example, fire suppression in the United States of America (USA) is a common practice (Parisien and Moritz, 2009; Marlon et al., 2012) while active fire suppression in most parts of Africa is not commonly practiced. Most fire suppression in Africa is an indirect byproduct of changes in land surface properties through grazing and fragmentation (Archibald, 2016) . Hence, we modified the simplistic approach suggested by Pechony and Shindell (2009), guided by the results presented in Sect. 5.1.1, to better match with observed fire activity at specific regions."

9. This is unrelated but a potentially large issue: The authors are not interested in crop-land fires, which makes sense given their model system's limitations, but why then do they not filter out MODIS hotspots and GFED emissions based on which were detected on cropland? One of the MODIS products is a land cover map (three, actually) that has been used in previous work to filter out hotspots detected on cropland, either for the purpose of discarding them or analyzing them on their own.

We are very interested in crop-land fires. The difference between crops and other vegetation is that crops are being burned intentionally with a specific seasonality that is sensitive to the crop type. This requires a dedicated study to implement in any model. Through another project we are working on improving the model's crop representation which would enable us to study cropland fires explicitly, which will include a retuning of active fires in the presence of both cropland and wildfires. We chose this general approach so we can imitate the total fire emissions.

10. It's also worth pointing out that the authors commit an all-too-common mistake in conflating MODIS hotspot detections with "fire counts." One large fire might have a fireline long enough for multiple hot pixels to be detected; a slow-moving fire might result in pixels that are counted multiple times as hotspots despite being

part of one fire. The authors' use of the hotspot data itself is not necessarily flawed—it's fine for the "fire counts" parameterization to target hotspots since it's all ideally going to get worked out in the emissions factors—but the authors should revise the manuscript to clarify exactly what it is the remote sensing data show. A true "fire counts" product is something more like the Global Fire Atlas (https://www.earth-syst-sci-data.net/11/529/2019/).

Thank you for this correction. We have added in L424 (revised version) the following: "One single fire might include multiple fire pixels." As the Referee mentions this is being worked out throughout our entire methodology, from scaling the simulated fire count to match with MODIS fire count pixels and through the calculation of emissions. Following this comment we have revised throughout the paper the use of "simulated fire count" to "active fires".

**Minor comments**

11. Some parts of the manuscript are too detailed and/or technical. For example, the authors spend over a page in the Introduction discussing the ways that people use fire to manage land, and exactly when those land management fires occur. For a process that's not even represented in the model, that seems like a pretty big waste of space. I would prefer to see relevant information along these lines presented instead in the Results and/or Discussion, to provide context for poor performance in some regions. (The authors do a bit of this, but more would be an improvement.)

We edited the introduction and removed most of the content of L98-133 (submitted version), instead of that part we now only include the following:

"Precipitation and fire activity are sensitive to natural modes of variability like El Niño Southern Oscillation (ENSO). In particular, the Southern Hemisphere BB activity is strongly coupled to ENSO [Buchholz et al., 2018]. During an El Niño year regional BB emissions can be up to two times higher than their regional average level, due to increased fire activity in tropical rainforests [van der Werf, 2004; Andela and Werf, 2014; Field et al., 2016; Whitburn et al., 2016].

Forest fires are either ignited on purpose, as part of forest management practices [Ryan et al., 2013], ignited by accident, as a by-product of the expansion of urban life to the wildland interface [Moritz et al., 2014; Fischer et al., 2016; Radeloff et al., 2018], or ignited by lightning [Díaz-Avalos et al., 2001]. Thus, fire activity is highly coupled to trends in population density as increased population density at the wildland-urban

interface (WUI) increases the probability of fire [Radeloff et al., 2018], while land abandonment leads to shrub encroachment, and fuels fire activity [Butsic et al., 2015]."

12. It is also unnecessary to get into the technical details of remote sensing, such as the sensor channels used to detect fire counts or the reflectance characteristics used in calculating burned area.

Following the recommendation of the Referee we edited Section 4 and removed L420-23, L443, L448-453, L462 (submitted version).

13. It seems that burned area is only used as an input to flammability in subsequent time steps. This should probably be made clearer, given that burned area is a primary product of most existing fire models. More importantly, though: How long does it take for that effect to fade? Is it just how much of the grid cell has burned EVER? Surely not, but the authors don't specify how this works.

We added in the model implementation section the following paragraph (L363-371, revised version): "All fire-related parameters like flammability, active fires, burned area, and fire emissions are recalculated in every model time step (30 min) with memory only of the burned area in the previous time step. We could not extend the "fire memory" past the previous time step due to limitations related to ModelE's terrestrial biosphere module. However this is a reasonable assumption, given that the climate inputs we use for fire calculations such as monthly accumulated precipitation, surface RH and temperature don't change significantly between each time step. The fire module's impact on the Earth system is currently only through interactive emissions. Albedo, carbon stocks and LAI are not modified by pyrE."

14. Why do the white (zero) areas on the right (model-simulated) side of Fig. 8 not match up exactly?

**We have fixed that, it was due to a plotting error related to contour interpolation.**

15. Figures throughout could use more labeling. It makes it hard to interpret figures when the reader needs to keep going back to the caption to figure out what's in the right vs. left column, top vs. bottom row, etc.

We have revised the figures and added labels.

16. I understand what the authors are getting at here given the context (at the end of the presentation of results of the simulation at times of day equivalent to the MODIS overpasses) but this sentence makes little sense and should be reworked. It sounds like "Even though A=B, A>B.": "The implications of these findings are that even though the simulated monthly mean fire count is in the range of Terra

**and Aqua (Fig. 4, A1), the simulated fire count is in fact higher than MODIS retrievals." [Lines 541–543]**

We edited the sentence to:

*"When simulated monthly mean active fires values are in the range of Terra and Aqua (Fig. 4, A1), they are in fact biased high, given the bias due to the overpass time of the satellite"*

17. These related sentences were extremely confusing until eventually I remembered about how previously-burned area affects flammability; this should be clarified:
- "Nevertheless, even with this large correction factor, burned area has a very minor impact on fire count and fire emissions as it accounts for a small fraction of the grid cell that is able to burn." (lines 382–384)

We edited the sentence to: "Nevertheless, even with this large correction factor, burned area, which accounts for a small fraction of the grid cell that is able to burn, has a very minor impact on fire activity and fire emissions as its only impact to fire activity is through flammability."

- "burned area itself has a minor impact on fires due to its small percentage in a grid cell" (lines 577–578)

We edited the sentence to: "burned area itself has a minor impact on fires through flammability due to its small percentage in a grid cell."

- I first noticed this at Line 588 and following, but it may have occurred earlier: The authors seem to sometimes incorrectly refer to the NHAF region as "sub-Saharan Africa." Sub-Saharan Africa in fact refers to ALL of Africa south of the Sahara, not just the northern-hemisphere portion.

We changed to "NHAF" or "northern sub-Saharan Africa" throughout the manuscript.

**Technical corrections** -

- 18. "Bias-high" and "bias-low" throughout should be "biased high" and "biased low".
- **19. Line 618: Should be cold- and drought-deciduous:** *it is cold broadleaf trees and drought broadleaf trees, not deciduous.*
- 20. Line 663: Extra period
- 21. Line 670: Missing word: replaced "using" with "with"
- 22. Line 677: Incorrect capitalization of pyrE
- 23. Line 684: Extra comma
- 24. Line 699: Missing comma at end of line

**25. Line 700: Should be Middle EAST, presumably**

26. Fig. A1 BONA: Truncated "1" at top of Y axis

All technical corrections above are now fixed throughout the text.

**Response to Referee #2:**

**General Comments:**

 The present paper presents a new fire module as implemented in the GISS ModelE ESM. The fire module relies on previous fire model developments. It is nice to see new developments in fire modeling, here focusing on fire emissions. Overall the manuscript is well written and the results presented clearly. PyrE does not seem to outperform existing fire models (which was clearly not the objective either), but its performance is neither bad compared to the performance of other fire models. My main issue with the current manuscript is that it is assuming fire numbers and active fire counts to be the same thing, which is confusing and can lead to wrong interpretation of results.

We have rephrased the manuscript to use the term "active fires" instead of "fire count", see reply #10 to Referee #1.

2. While I don't have any problem with the chosen approach of moving directly from fire count to emissions, the authors should present their methods in a more coherent way.

Following the comments from Referee #1 we expanded how this was performed. Please see the reply to comment #7 from Referee #1.

Below are some comments which I hope will explain in more detail where I think the issues are.

**Detailed comments:**

3. L35: pyrE: Is this an acronym? If so, indicate this.

In L226 (submitted version) we explain that pyrE comes from pyr the Greek word for fire. We added the Greek spelling to the revised version.

- 4. L40: "...as is commonly done in other interactive fire modules. ", this is not true. We edited this statement, see reply #1 to Referee #1.
- 5. L98-123: It is unclear to me why you have here such a huge emphasis on regional seasonality of fires. It distracts a bit from the main message you want to bring here, which is presented in the following section where you do explain the

reasons why you are so much interested in incorporating a fire module within your ESM.

We significantly reduced this part of the manuscript. Please see reply #11 to Referee #1.

- 6. L216: "fire counts" are not really a much-used term within the fire community, thus making it hard to understand what you mean. Use "number of fires" or similar to avoid confusion. Fire counts could be interpreted as thermal anomaly detections from satellites etc., which is something completely different. See reply to comment #1. We would also like to note that the FireMIP paper by Rabin et al., (2017) as well as previous work by Pechony and Shindell which is the base of our study use the term fire count.
- 7. L220-223: while I think I know what you want to indicate here (flaming fase compared to area burnt), I don't agree with this. While this could be a computationally faster way (which can be an argument), there are also inconveniences associated, with e.g. no fuel gradient driving your emission amounts within each region. Again, I don't have anything against your approach, but you should explain things clearer to avoid confusion. See reply #2 to Referee #1.
- 8. L370-373: may add this to section 4 and change that name to "datasets", as it has nothing to do with ModelE.

Agreed. Changed Section 4 to "4. Dataset" and "4.1 Population density" with the text from L370-373 (submitted version) within excluding the part from the Referee's following comment.

- L372: "and on future projections (not used in this study) for years past 2010 ". If it is not applicable here, no need to mention it. *Agreed, deleted.*
- 10. L375-379: This is not really accurate, as some models here just assign a fixed fire size instead of a dynamic one, others are completely empirical and hence don't simulate fire numbers etc.

The papers we cite here are process-based fire models that use scaling factors in a similar manner to our model.

11. L379-380: This is confusing: during the methods you indicate that you produce number of fires and that you incorporate a fire growth method to obtain burnt area. Hence your fire counts = fire numbers. However, here it seems to indicate that the authors of the manuscript use two completely different concepts "fire number" and "satellite observed thermal anomalies" indiscriminate while they cannot be compared one-to- one. Thermal anomalies/active fire counts are detections of whether within a pixel at the time of the satellite overpass there is an fire actively burning (or smouldering), so you can have 100s or 1000s of active fire detections within 1 single fire (and hence number of thermal anomalies is closely related to burnt area and not number of fires at a regional scale). So, one would expect to have more active fires than number of fires, and hence the need of a scaling factor. This is a problem within this manuscript and should be fixed. As you need an even much larger scaling factor for burned area it mainly seems that your fires don't grow large enough

See reply #10 to Referee #1.

- 12. As a side note, and again, no problem if you ignore this. Have you thought about ignoring burnt area completely, as it barely used (no terrestrial impact as far as described in the manuscript) and doesn't seem to influence your results much? Though the simulated burned area is far from perfect, it will be developed in the future, when crop fires will be included. We strongly feel that the development work thus far needs to be documented and prefer to keep it in the paper.
- 13. L403: why 2005 and not a climatology as well ? Now you focus on a specific year which its peculiarities (El Niño with much fire in the amazon).
   2005 Climatological GFED emissions are based on an interpolation of climatological

emissions from 2000 and 2010. We edited L397-398 to clarify that: "using prescribed 2005 climatological (interpolated 2000-2010) GFED4s emissions".

14. L409: This is personal, but I find this section way to detailed for just describing some of the reference datasets used. Especially considering the brief introduction to modelE, which I think is more important (at which time resolution do you run?), Especially the integration of the fire module within the land surface model. Does the fire model change albedo? Carbon stocks? LAI? I don't think these important parts are covered, and would prefer that compared to these detailed descriptions of external datasets.

As stated in reply to comment #12 to Referee #1 we have deleted some technical parts of Section 4. The details on the implementation and simulation were extended, in response to comment #13 of Referee #1. Also, we now added in L369-371 (revised version): "The fire module's impact on the Earth system is currently only through interactive emissions. Albedo, carbon stocks and LAI are not modified by pyrE."

15. L471: GFED4s starts in 1997 (1995 is mentioned a couple of times more in the manuscript)

Corrected.

16. L522: This is great and the first time I see (output) of sub-daily fire model results. I think it would be nice that it is indicated in the methods as well. So, this is completely driven by changes in VPD, which seems indeed to capture the expected sub-daily dynamics surprisingly well. Have you thought of comparing this to GFED 3h estimated emissions or thermal anomalies from some geostationary satellite? Just a thought, so don't feel obliged.

The standard model output is only saving monthly mean diagnostics and to extract output in finer resolution would require to repeat simulations which is something that is not easy for us to do right now. However, it is a great idea and we would strongly consider it for future applications of this model.

17. L554-555: I guess you divided these numbers by accident by 10. GFED4s burnt area is somewhere around 450Mha (did you use GFED4 instead which is around 350Mha?)

We thank the referee for noticing this glitch. We had erroneously presented the monthly mean BA for the 2003-2016 time period instead of the annual mean (a factor of 12 error). We have corrected the text with the value of 460Mha (the 2003-2016 mean is in fact 457Mha). Following this we have also corrected the text with the value of 380Mha for the simulated annual burned area (L544-L545 revised version).

- 18. **Table 2: Indicate time period.** *Added to the table title: "Modeled annual emissions and column load means are based on an ensemble of 10 simulations. GFED4s emissions are based on a 2000-2010 climatological mean."*
- 19. Figures: increase fond size of most figures, as many legends and axis are now barely readable.

We have increased the font size in Figures 5, 7-12 and also added alphabetic labels to Figures 4-5, 7-12.

20. Figure 3 can go to supplementary, as burnt area is anyway not the main focus of the fire module.

The paper includes a full section on burnt area and describes in detail the challenges related to simulating it. Though it is not a key metric for pyrE, as it does not drive emissions, it is still a benchmark metric for fire models and we think it is appropriate to keep it in the main paper.

21. Figure 5: why didn't you indicate the values for Terra and Aqua instead of a star. If you don't want to show the values, I think the stars are a bit misleading as they are put straight on the mean model output, so you better draw a shaded box, line or something similar.

Since the plot includes the diurnal cycle averaged over a month it would be messy and confusing to overlap the variability bars with a box that represents the monthly mean values of Terra and Aqua. This particular (CMG) MODIS product is not available in daily resolution. We tried to change the location of stars but found it to look even more confusing and thus prefer to keep the plot as is.

22. Figure 7-9: indicate which map indicates what, be it with letters or a description above each subplot.

Added alphabetic labels to Figures 4-5, 7-12.

**Response to Referee #3:**

1. This paper describes the updates of pyrE, the fire component of GISS ModelE. The discussion of the paper is written very well, demonstrating the comprehensive grasp of fire modeling by the authors. If this were the first paper describing pyrE, I would firmly support its publication. However, 10 years after the papers by Pechony and Shindell, the current updates of pyrE do not appear to have sufficient improvements to justify its publication. My guess is that the updates of pyrE proved to be more difficult than the researchers anticipated. As an important component of the venerable ModelE, pryE updates obviously need to be documented. I wonder if it would be better to include pryE updates as part of another ModelE paper rather than a standalone paper.

This is the first paper describing pyrE, a significant advancement since the Pechony and Shindell work, and we strongly believe that a full paper dedicated to pyrE is necessary. As mentioned by Referee #1: "pyrE builds upon work by previous authors but includes some novel elements that could be of interest to the broader community of global fire modelers". Pechony and Shindell (2009) developed a fire parameterization that was driven by input from previous versions of ModelE but was never used interactively within ModelE. Our paper documents the full implementation of interactive fire emissions in ModelE, including some undocumented features of the past model. The additional dependencies include:

-Regional fire suppression (briefly described in L200-L202 (revised version) -Fire spread and burned area (briefly described in L209-210 (revised version) -New emission factors based on GFED4s emissions L339-L346 (revised version)

 In model description, sections 2.1, 2.2, 2.4, and probably some of 2.5 are essentially the same as Pechony and Shindell (2009). For a journal paper, these sections should be summarized in 1-2 paragraphs and the authors can refer readers to Pechony and Shindell (2009).

A fully detailed description of the model is appropriate for a paper describing the parameterization and we prefer to keep it in the paper. We already cite the original work where appropriate.

- 3. Section 2.5 is new. However, burned area (BA) modeling is odd in this paper. The purpose of developing it is unclear. The introduction section is 7 pages long, but it did not include discussion on why modelE needs to simulate BA when fire emissions are calculated using fire counts. This is not a review paper. The introduction section is too verbose and should be shortened substantially. *We reduced the length of the introduction following comment #11 from Referee #1 and comment #14 from Referee #2*
- 4. The BA modeling is based heavily on Li et al. (2012) and the biases compared to GFED4s are quite large (Figure 8). It seems to be worse than the other global fire models cited in the paper. If BA modeling does not serve a useful purpose for modelE, it should be removed.

See reply #12 to Referee #2.

5. I am alarmed by section 2.7. Line 368 states lightning ignition is scaled down by a factor of 10, but line 380 states that fire counts are scaled up by a factor of 30. These statements do not inspire confidence in modelE simulations. pyrE is supposed to be a physical model. Does this large a scaling factor imply that the model is not really physical? How is 30 determined? With such a large arbitrary scaling factor as a tuning knob, why is the scaling factor not tuned such that the simulated global fire counts agree with satellite observations (rather than a 32-42% low bias)?

pyrE is not a physical model, it is a process-based parameterization, as stated in L215, L229-230 submitted version. To clarify we edited L210-212 (revised version):

"The module is a collection of physical processes like flammability, natural ignition, fire spread, and fire emissions, and empirical processes that include accidental ignition and suppression (Fig. 2)."

pyrE, like many other fire models, tries to incorporate physical processes by using physical and empirical parameterizations from the literature. The parameters include coefficients in the Goff and Gratch equation of vapor pressure, an empirical constant related to the inverse of precipitation at the Keetch and Byram equation, anthropogenic ignition and suppression coefficients, fire spread and burned area coefficients. These empirical parameters are probably model-dependent, but instead of modifying those we chose to keep them as-is and apply the above-stated scaling factors. This is a common practice in models of all scales. We have performed a series of sensitivity simulations to find the best global fit.

 One can go a step further to tune this factor differently for each region. For modelE, it seems that a statistical fire model would work better than the current setup.

Tuning per region is better to represent a given reality but we wanted to have a model that could be applied in different regimes from paleo to future and adding the spatial distribution of present day would make the model less applicable to other time periods.

- 7. Line 381 states that BA is scaled up by a factor of 250. After discounting the factor of 30 in fire count scaling, BA is still scaled up a factor 8. Li et al. (2012) did not have to do this large scaling in their model. What went wrong here? Nothing went wrong, it's a different model as stated in responses #5 and #6. A parameterization from one model doesn't work out of the box in another model.
- 8. Sections 2.3 and 5.1.1 are new improvements to pyrE. Discuss how Eq. (6) was derived. Is it through some kind of linear regression?

We added to the text the following (L275-L281, revised version): "Our initial analysis showed that with the original Pechony and Shindell (2009) suppression scheme fire activity is overestimated in the TENA and MIDE regions while being underestimated in NHAF and SHAF. Following these initial results a series of sensitivity simulations included varying values of suppression coefficients. The final values were chosen in a heuristic manner that improved the simulations yet did not overfit them to the observations, similarly to Pechony and Shindell (2009) and other fire parameterization, due to the lack of appropriate global data. "

The exact values of the coefficients, including in regions other than TENA, MIDE, NHAF and SHAF, need to be addressed in future research.

9. The evaluation in section 2.6 is inadequate. Fire emissions are what matter to modelE simulations. Regional emission biases of pyrE OA and BC relative to GFED4s should be discussed. The column load comparison in Table 2 is not informative and should be removed. If fire emission is underestimated but the column load is not, it only shows that fire emissions are unimportant (and pyrE is not needed). Fire emissions may not significantly change global mean column loads but they strongly affect the atmosphere, which the authors described in the introduction.

Regional fire emissions are discussed in Section 5.3 (Emissions) of the Results and Discussion part of the paper. They are compared and evaluated against GFED4s emissions. E.g. in L581-L591 (submitted version): "Emissions are well simulated over SHSA and SHAF (Fig. A3-A5), both in terms of timing of the seasonality and in magnitude. The main regions where simulated emissions are lower than GFED4s are NHAF and EQAS, mainly Indonesia (Fig. 8, A3-A5)."

We think that the discussion about the column load is informative as this is the step that connects fire emissions to atmospheric composition and climate. We agree that fires' impact on composition is mostly regional and closer to the surface, which is why we include Figure 9 in the paper. However, similar to the emissions in Figure 8, we think it is fitting to include the information on the global mean values.

We added in L626-L627 (revised version) the following: "Having a weak global impact on composition does not imply that regionally fires are not important." to introduce the following paragraph that discusses the regional differences.

10. In the abstract, line 40-42 states "Using pyrE, we examine fire behavior, regional fire suppression, burned area, fire emissions, and how it all affects atmospheric composition." In the current setup, fire behavior and burned area do not affect atmospheric composition in modelE.

We have edited the "fire behavior" to "fire occurrence" in the abstract, but not burned area as it does affect composition, though marginally.

11. In the abstract, the last sentence, "Yet, in terms of AOD, a simulation with interactive fire emissions performs just as well as simulation with prescribed fire

emissions", is misleading. It sounds as if global fire emission biases as large as 40% do not matter for modelE simulations, which I think is not what the authors meant. The pyrE and GFED4s simulations of AOD are very different in regions where fire emissions are present and large.

We agree and have rephrased this sentence:

"Regionally, the resulting AOD of a simulation with interactive fire emissions is underestimated mostly over Indonesia compared to a simulation with GFED4s emissions and to MODIS AOD. In other parts of the world pyrE's performance in terms of AOD is marginal to a simulation with prescribed fire emissions."

12. Figure 12 shows that fire AOD effect is only about 10% (line 676-677). The AOD evaluation in this paper should be about fire emissions not modelE AOD simulation. Figures 9 should compare fire-only AOD between pyrE and GFED4s simulations. The differences are large. Figure 11 is not useful because the non-fire AOD model biases are much more apparent than fire AOD.

Our model configuration uses internally mixed aerosols which makes it impossible to compare the BB-only AOD.